# Finetuned Super-Resolution Generative Adversarial Network (Artificial Intelligence) Model for Calcium Deblooming in Coronary Computed Tomography Angiography

**DOI:** 10.3390/jpm12091354

**Published:** 2022-08-23

**Authors:** Zhonghua Sun, Curtise K. C. Ng

**Affiliations:** 1Discipline of Medical Radiation Science, Curtin Medical School, Curtin University, Perth, WA 6845, Australia; 2Curtin Health Innovation Research Institute (CHIRI), Faculty of Health Sciences, Curtin University, Perth, WA 6845, Australia

**Keywords:** calcification, coronary computed tomography angiography, deep learning, generative adversarial network, plaque

## Abstract

The purpose of this study was to finetune a deep learning model, real-enhanced super-resolution generative adversarial network (Real-ESRGAN), and investigate its diagnostic value in calcified coronary plaques with the aim of suppressing blooming artifacts for the further improvement of coronary lumen assessment. We finetuned the Real-ESRGAN model and applied it to 50 patients with 184 calcified plaques detected at three main coronary arteries (left anterior descending [LAD], left circumflex [LCx] and right coronary artery [RCA]). Measurements of coronary stenosis were collected from original coronary computed tomography angiography (CCTA) and Real-ESRGAN-processed images, including Real-ESRGAN-high-resolution, Real-ESRGAN-average and Real-ESRGAN-median (Real-ESRGAN-HR, Real-ESRGAN-A and Real-ESRGAN-M) with invasive coronary angiography as the reference. Our results showed specificity and positive predictive value (PPV) of the Real-ESRGAN-processed images were improved at all of the three coronary arteries, leading to significant reduction in the false positive rates when compared to those of the original CCTA images. The specificity and PPV of the Real-ESRGAN-M images were the highest at the RCA level, with values being 80% (95% CI: 64.4%, 90.9%) and 61.9% (95% CI: 45.6%, 75.9%), although the sensitivity was reduced to 81.3% (95% CI: 54.5%, 95.9%) due to false negative results. The corresponding specificity and PPV of the Real-ESRGAN-M images were 51.9 (95% CI: 40.3%, 63.5%) and 31.5% (95% CI: 25.8%, 37.8%) at LAD, 62.5% (95% CI: 40.6%, 81.2%) and 43.8% (95% CI: 30.3%, 58.1%) at LCx, respectively. The area under the receiver operating characteristic curve was also the highest at the RCA with value of 0.76 (95% CI: 0.64, 0.89), 0.84 (95% CI: 0.73, 0.94), 0.85 (95% CI: 0.75, 0.95) and 0.73 (95% CI: 0.58, 0.89), corresponding to original CCTA, Real-ESRGAN-HR, Real-ESRGAN-A and Real-ESRGAN-M images, respectively. This study proves that the finetuned Real-ESRGAN model significantly improves the diagnostic performance of CCTA in assessing calcified plaques.

## 1. Introduction

Coronary artery calcium scoring is widely used in patient screening to enable a more personalized risk assessment [1,2,3]. However, blooming artifact of coronary computed tomography angiography (CCTA) resulting from extensive calcification within the coronary plaques affects the accurate assessment of coronary stenosis, thus leading to high false positive rate. “Blooming” in the calcified plaques refers to partial volume averaging of different densities within a single voxel in the coronary arteries, and this is usually caused by limited spatial resolution of computed tomography (CT) scanners. High-density calcium overwhelms the attenuation of other tissues in the voxel and adjacent structures, and thus exaggerates the dimension of the highly calcified plaque (Figure 1). Hence, the high-density calcified plaque appears larger than it is or is “bloomed”, which negatively affects the visualization and assessment of the coronary artery lumen and the degree of stenosis. The consequence of blooming artifact is the overestimation of the coronary stenosis, which compromises the specificity and positive predictive value (PPV) of CCTA but does not change the sensitivity of CCTA. This leads to unnecessary downstream testing, usually invasive coronary angiography (ICA), which should be avoided in patients without significant coronary stenosis [3,4,5,6].

One of the main approaches for blooming artifact suppression is to improve the CCTA image spatial resolution. Various strategies for increasing the CCTA image spatial resolution to reduce the blooming artifact have been reported [3,4,5,6]. The latest strategy is to use artificial intelligence (AI) (specifically deep learning [DL]), including convolutional neural network (CNN)-based CT image reconstruction kernels, such as Canon Medical Systems Advanced Intelligent Clear-IQ Engine (AiCE) and generative adversarial network (GAN) model, for image postprocessing, to achieve this goal [3,7].

Our recent study has shown that enhanced super-resolution GAN (ESRGAN) was able to effectively suppress the CCTA blooming artifact and improve the specificity and PPV by 10–40% for patients with heavy calcification in the coronary arteries [3]. Its performance was better than the Canon AiCE reconstruction kernel, which could only increase the PPV by about 10% [3,7]. Despite these promising results, AI inference was used in our previous study, i.e., no medical image was used to train the ESRGAN model for the calcium deblooming task [3,8,9]. Hence, one straightforward way to further improve the performance of the ESRGAN model for this task is to finetune the model with use of CCTA images [3,8,9,10,11]. The use of finetuning (a subset of transfer learning) has become popular in the medical imaging field because of limited availability of medical images for training a model from scratch and it being time- and resource-efficient but still being able to achieve superior performance on similar tasks [12]. The purpose of this study was to finetune the ESRGAN model with the use of CCTA images and evaluate performance of the finetuned model on calcium deblooming in CCTA. We hypothesized that the use of the finetuned model would further improve the coronary artery stenosis assessment on the CCTA images with heavy calcification in the arteries and hence the calcified plaque diagnosis. With further improved diagnostic value, AI-assisted CCTA will lead to reducing false positive rates, thus contributing to the reduction in unnecessary downstream examinations, such as ICA procedures.

## 2. Materials and Methods

### 2.1. ESRGAN Model Finetuning

The open-source, pre-trained ESRGAN (known as Real-ESRGAN) model by Wang et al. was used in this study. Its source code in PyTorch v1.7.0 (Meta Platforms, Inc., Menlo Park, CA, USA) was available at https://github.com/xinntao/Real-ESRGAN (accessed on 26 April 2022) [13]. The Real-ESRGAN model was an enhanced version of the ESRGAN model reported in our previous study. For example, U-Net design was used in the discriminator of the Real-ESRGAN model. [3,13,14]. Details of the enhancement were available from Wang et al.’s article. Although the same publicly available datasets used for training the original ESRGAN, i.e., DIV2K (https://data.vision.ee.ethz.ch/cvl/DIV2K/ (accessed on 1 February 2022)), Flickr2K (http://cv.snu.ac.kr/research/EDSR/Flickr2K.tar (accessed on 1 February 2022)) and OutdoorSceneTraining (OST) (http://mmlab.ie.cuhk.edu.hk/projects/SFTGAN/ (accessed on 1 February 2022)) with a total of 13,774 non-medical images were used for training the Real-ESRGAN model with 400,000 iterations, these images were sharpened before using them as ground-truth images for the training [13].

For finetuning Wang et al.’s Real-ESRGAN model to achieve better performance on calcium deblooming [13], 32 deidentified CCTA datasets acquired by a 640-slice CT scanner (Toshiba Aquilion ONE, Toshiba, Otawara, Japan) in 2015 with a reconstruction slice thickness of 0.5 mm and interval of 0.25 mm in Digital Imaging and Communications in Medicine (DICOM) format of patients with heavy calcification in the coronary arteries were collected. Institutional review board approval was waived and informed consent was not required as the nature of this study was retrospective and the CCTA procedure was part of the diagnostic process. The 32 datasets consisted of 16,904 images with an equal size of 512 × 512 pixels. All datasets were used to train both generator and discriminator of the Real-ESRGAN model with 40,000 iterations through the free Kaggle platform (Google LLC, Mountain View, CA, USA) with one NVidia K80 graphics processing unit (Santa Clara, CA, USA). The finetuning process took about 259 h to complete. Forty thousand iterations were used because Wang et al.’s study showed that this setting was more than adequate to finetune a pre-trained GAN model to carry out a similar task with an optimum performance [11]. 

### 2.2. Finetuned Real-ESRGAN Model Performance Evaluation

The finetuned Real-ESRGAN model was used to postprocess 50 CCTA datasets not involved in the finetuning process, i.e., to increase the image spatial resolution from 512 × 512 pixels (original resolution) to 2048 × 2048 pixels (high-resolution) for calcium deblooming. Approximately 10 min was needed to postprocess each dataset (hundreds of images) with the use of the Kaggle platform. These 50 CCTA datasets with paired reference images (ICA datasets) were those used in our previous study for evaluating the performance of the ESRGAN model on blooming artifact suppression. Our previous approach to evaluate the ESRGAN model was also adopted in this study. Details of these datasets and evaluation strategy were available from our previous article. The following is the summary of our evaluation approach and Figure 2 illustrates the key steps involved in the Real-ESRGAN model finetuning and its performance evaluation [3].
Generation of two other CCTA datasets with an image size of 512 × 512 pixels based on the high-resolution images (Real-ESRGAN-HR) (2048 × 2048 pixels) through average (Real-ESRGAN-Average) and median (pixel) binning (Real-ESRGAN-Median) approaches for further image noise reduction.Measurements of minimal lumen diameter (MLD) at each calcified plaque lesion of three main coronary arteries, left anterior descending (LAD), left circumflex (LCx) and right coronary artery (RCA) for the 200 datasets (50 original CCTA, 50 Real-ESRGAN-HR, 50 Real-ESRGAN-Average and 50 Real-ESRGAN-Median datasets) by a single researcher (with experience of more than 20 years in CCTA image interpretation) for three times per lesion with average value taking as the final. The MLD was measured at the narrowest part of each coronary lumen (the most extensively calcified area) to determine the degree of stenosis on the original CCTA and Real-ESRGAN-processed images with measurements on ICA as the reference to calculate the diagnostic value.Determination of blooming artifact reduction by using Formula (1) below.
[(MLD_Real-ESRGAN-Processed Image_ − MLD_Original CCTA Image_) / MLD_Original CCTA Image_] × 100%(1)

**Figure 2 jpm-12-01354-f002:**
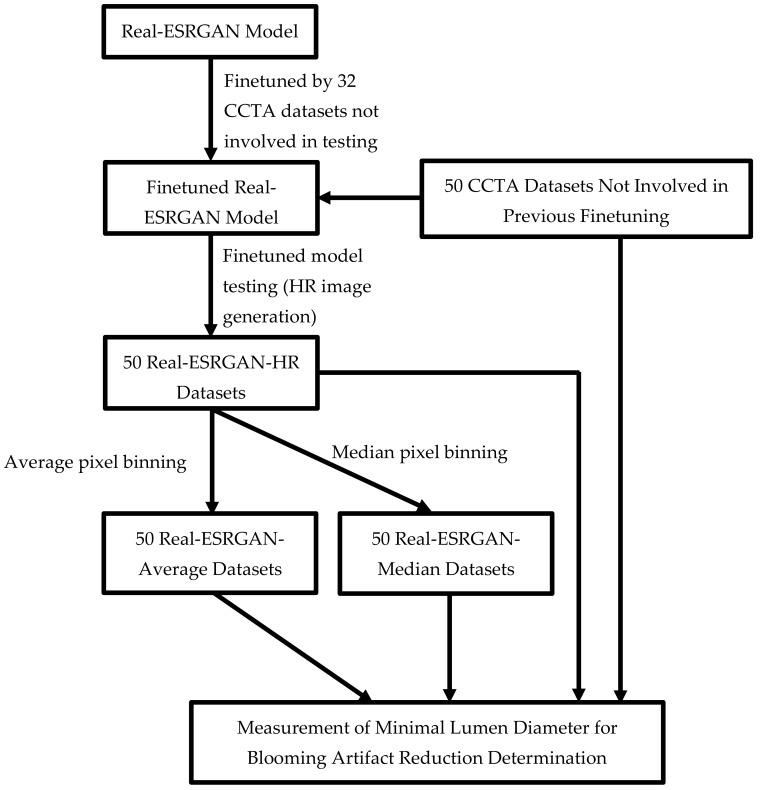
Flowchart showing key steps involved in real-enhanced super-resolution generative adversarial network (Real-ESRGAN) model finetuning and its performance evaluation. CCTA—coronary computed tomography angiography; HR—high resolution.

### 2.3. Statistical Analysis

IBM SPSS Statistics 27 (New York, NY, USA) was used for statistical analysis. Mean ± standard deviation (SD) and percentages were used for presenting continuous and categorical variables, respectively. For the 50 original CCTA, 50 Real-ESRGAN-HR, 50 Real-ESRGAN-Average and 50 Real-ESRGAN-Median datasets, sensitivity, specificity, PPV, negative predictive value (NPV), positive likelihood ratio (PLR) and negative likelihood ratio (NLR) were calculated and compared across these four groups with ICA as the reference. The ICA measurements were conducted in our previous study and their details were available from our recent paper [3]. Diagnostic performances of these 4 groups were determined through receiver operating characteristic (ROC) analysis. The measurements from the 5 groups (original CCTA, Real-ESRGAN-HR, Real-ESRGAN-Average, Real-ESRGAN-Median and ICA) were compared through three-way analysis of variance (ANOVA) with post-hoc pairwise comparisons. A *p*-value less than 0.05 indicated statistical significance.

## 3. Results

There were a total of 184 calcified plaques that were assessed in this study with the same plaque distribution at these three main coronary arteries as reported in our recent study [3]. Compared to the measurements on ICA, original CCTA and Real-ESRGAN-processed images overestimated the degree of coronary stenosis resulting in significant differences in coronary lumen measurements (*p* < 0.01), as shown in Figure 3.

We randomly selected 20 cases for testing intra-observer reproducibility of measurements among the original CCTA, Real-ESRGAN-processed images and ICA with an interval of 8 weeks between the first and second measurements. High correlation was achieved in MLD values by the same observer among all the measurements (*r* = 0.937–0.991, *p* < 0.001).

Of these Real-ESRGAN-processed images, the Real-ESRGAN-Median images showed the most significant improvements in the degree of reducing blooming artifacts, with the mean value and SD being 10.99 ± 13.94%, 14.42 ± 14.78% and 18.06 ± 15.74% at LAD; 14.57 ± 10.13%, 17.53 ± 9.64% and 22.02 ± 12.02% at LCx; 14.74 ± 11.90%, 16.63 ± 12.01% and 23.81 ± 14.96% at RCA, corresponding to Real-ESRGAN-HR, Real-ESRGAN-Average and Real-ESRGAN-Median images, respectively. Figure 4 shows the percentage of the reduction in coronary lumen measurements at the three main coronary arteries, as assessed by these three Real-ESRGAN-processed images when compared to those from the original CCTA images. Although the Real-ESRGAN-Median images led to the highest degree of reduction in most of the plaque assessments (>90%), indicating its significant impact on suppressing the blooming artifact, increased overestimation of the coronary lumen (by 11–32%) was observed in seven plaques at LAD (plaque numbers 23, 25, 30–33 and 52), when compared to the original CCTA and other Real-ESRGAN-processed images. In contrast, this phenomenon was not observed at LCx and in only two plaques (plaques numbers 27 and 36) at RCA (Figure 4).

The number of false positive rates was found highest in the original CCTA images, resulting in the lowest specificity and PPV at all three coronary arteries as shown in Table 1. The number of false positive rates was reduced when applying the Real-ESRGAN model to postprocess the original CCTA images, with Real-ESRGAN-Median images showing the significant impact on reducing the false positive rates. The specificity and PPV were significantly improved with Real-ESRGAN-Median images, compared to original CCTA, Real-ESRGAN-HR and Real-ESRGAN-Average images at all three coronary arteries (Table 1). With use of Real-ESRGAN-Median images, the specificity and PPV achieved 80% and 61.9% at RCA, 52–62% and 32–44% at LAD and LCx, respectively, although false negative cases were found in the Real-ESRGAN-processed images, which decreased the sensitivity to some extent. The area under curve (AUC) of ROC analysis was higher in Real-ESRGAN-processed images than that in the original CCTA, as shown in Figure 5. The highest AUC was found in Real-ESRGAN-processed images at RCA level (Table 1).

Figure 6 is an example of multiple calcified plaques at LAD with improved visualization of coronary lumen observed in Real-ESRGAN-processed images, while Figure 7 is another example showing the calcified plaques at LAD with Real-ESRGAN-processed images resulting in false negative finding, when compared to original CCTA and ICA images.

## 4. Discussion

This study further advances our recent report of using the finetuned DL model, Real-ESRGAN, to postprocess the original CCTA images with results showing significant improvements over previous studies [3,4,5]. Based on analysis of the same dataset, our results showed that the specificity and PPV were further increased by up to 25% and 15%, respectively compared to our recent results [3], indicating that the finetuned Real-ESRGAN model allows for further improvement in assessing calcified coronary plaques. This has significant clinical impact as the number of false positive rates were reduced, thus reducing the unnecessary downstream testing, such as avoiding ICA procedures, when diagnosing calcified coronary plaques.

The assessment of coronary artery disease, particularly coronary calcification and coronary plaques, is a well-recognized issue in CCTA, which has drawn increasing attention in recent years to tackle this challenging area. Although a number of strategies have been implemented with some promising results, the use of an AI algorithm to process original CCTA images represents the most promising strategy in the recent literature [15,16,17,18,19,20,21,22,23]. Studies have shown that an AI algorithm allows for accurate and efficient quantification of coronary calcium scores and assessment of coronary stenosis, when compared to the standard manual approach or semi-automatic method [15,16,17,18,19]. AI is also shown to achieve good accuracy in characterizing plaque morphology and differentiating plaque from no plaque or calcified from non-calcified plaques [20,21,22,23]. However, very limited research has been conducted so far with use of AI in suppressing heavy calcification in the coronary arteries for reducing blooming artifacts to improve lumen assessment. Further, most of the previous studies used the traditional CNN model, which is inferior to the advanced GAN approach. Most applications using the GAN approach focus on CT denoising [24], coronary artery disease risk categorization by quantifying calcium scoring [25] and automated registration of positron emission tomography-CT angiography images in imaging coronary artery disease [26]. Our study has addressed this gap by using the latest GAN model with promising results achieved.

Inage et al. [27] in their recent study applied the cycle GAN-based lumen extraction model to CCTA images in 99 patients involving 891 segments with severe calcification in the coronary arteries. The diagnostic value of assessing coronary stenosis by the original CCTA and cycle GAN-processed images were compared with ICA as the reference method. In addition to assessing the performance of the original CCTA and cycle GAN-processed images in all 891 segments, authors focused on the analysis of 228 segments, which were not assessable on the original CCTA images due to severe calcification. Their results showed similar specificity and PPV between the original CCTA and cycle GAN-processed images (75.1% and 40.9% vs. 77.3% and 43.4%) among assessment of all coronary segments, with AUC significantly higher in the cycle GAN group than the original CCTA group (0.77 vs. 0.75, *p* = 0.03). For the non-assessable 228 segments, the cycle GAN model significantly improved the specificity and accuracy, compared to the original CCTA (10.9% and 42.5% vs. 0% and 35.5%, *p* < 0.001), along with significantly higher AUC (0.59 vs. 0.50, *p* < 0.001). Similar to assessment of all segments, the PPV was similar between these two groups (35.5% and 38.2% for the original CCTA and cycle GAN). Authors claimed that the use of cycle GAN model could avoid 4 out of 99 ICA examinations based on their study, with an estimated 747 ICA procedures to be avoided per year. Despite promising results achieved within that study, the PPV was low (<45%) with moderate specificity and very low specificity in all segments and non-assessable segments groups. In contrast, our findings showed much better results than those from Inage et al.’s study [27]. Both specificity and PPV were increased significantly with the use of finetuned Real-ESRGAN model, achieving 80% and 61% at the RCA level, although still low at LAD and LCx levels. This represents, so far, the most promising outcomes of diagnosing calcified plaques. Although we did not evaluate the economic benefits in our study, the improved PPV with reduced false positive rates will lead to avoidance of more unnecessary ICA examinations.

With further reduction in false positive rates leading to improved specificity and PPV with the use of finetuned Real-ESRGAN model compared to our previous report [3], the negative effect is the slightly decreased sensitivity due to the false negative rates. Up to three false negative cases were noticed in the Real-ESRGAN-processed images at all of three coronary arteries with the highest number seen in the Real-ESRGAN-Average and Real-ESRGAN-Median groups (Table 1) but not in the original CCTA images. This issue could be attributed to two factors. Firstly, only 32 datasets consisted of 16,904 CCTA images were used to finetune the Real-ESRGAN model. According to Wang et al.’s [11] study about transfer learning of pre-trained GAN models, this arrangement should be sufficient. The unexpected false negative cases would be an indication of the Real-ESRGAN model not exposed to adequate variations of plaque characteristics, e.g., varied compositions with presence of mixed calcium and atheromatous plaques, etc. More cases with greater varieties should be used to further finetune the model. Secondly, the use of average and median (pixel) binning was for reducing the noise presented within the Real-ESRGAN-HR images, leading to the enhancement of the visualization of coronary lumen for more accurate assessment. However, the pixel binning decreased the spatial resolution of Real-ESRGAN-Average and Real-ESRGAN-Median images by using the average and median values of four pixels to represent one pixel, respectively. Since the blooming artifact is caused by using the mean attenuation value of a calcified plaque with high density and a vessel with much lower density to represent these two objects, it was expected that the average binning would have lower performance than the median binning [3]. Our results were in line with this expectation except the aforementioned cases, which could be due to presence of mixed calcium and atheromatous plaques. Further finetuning of the Real-ESRGAN model with a greater number and variety of cases should address this issue.

One of the important themes in modern health care is personalized medicine, which generally refers to tailoring service delivery based on patient’s conditions. Our finetuned Real-ESRGAN model significantly improves the diagnostic performance of CCTA in assessing calcified plaques, which is one of the causes of coronary artery stenosis. Hence, our work advances the development of personalized medicine by using the latest DL technology to provide a better diagnostic service to a specific group of patients with the coronary artery stenosis caused by the calcified plaques [28].

Spatial resolution is one of the important elements for visualization of fine details in medical imaging, which is essential in accurate diagnosis of various pathological conditions. Hence, the use of Real-ESRGAN model can be extended to other related areas, for example, textual detail restoration for low dose CT images [29], visualization of small soft tissue foreign bodies on digital radiographs [30,31], etc. Nonetheless, the Real-ESRGAN model should be finetuned with relevant medical images before the applications. Another benefit of extending the use of Real-ESRGAN model in these areas is that GAN is less likely affected by the overfitting issue because the generator of the GAN model learns directly from its discriminator’s feedback instead of training / finetuning images. Hence, more robust performance would be expected [32]. 

This study has some limitations. First, although significant improvements in specificity and PPV were achieved over previous studies [3,4,5,7,27], the diagnostic values of the finetuned Real-ESRGAN-processed images at LAD and LCx are still low to moderate, particularly at the LAD level, since it has the largest number of calcified plaques. Further improvement of the Real-ESRGAN model is necessary to address this limitation. Second, as highlighted in our previous study [3], we did not investigate the diagnostic performance of the finetuned Real-ESRGAN model in differentiating calcified from non-calcified plaques as we focused on the heavy calcification in the coronary arteries, since this is the main challenging issue to be resolved with CCTA images. Although only 50 patient cases were included in this study but 184 plaques were analyzed, which was a greater number than the one of a similar study [7]. Moreover, for studies about use of AI in radiology, usually, about 50 patient cases were collected for clinical evaluation of the AI models [29]. With improved specificity and PPV, and high AUC achieved with our finetuned model, the use of the Real-ESRGAN model is expected to apply to large datasets with inclusion of cases with different types of plaques. Third, these 50 cases were scanned with different types of CT scanners (64-slice and beyond) with sufficient image quality achieved for the diagnostic assessment of coronary plaques. However, the heterogeneity of the original datasets could impact the AI-processed images, thus affecting final outputs of the image assessment. Ideally, CT imaging data from the same type of CT scanners should be used to avoid this issue and this will be addressed in our further study with inclusion of large datasets. Finally, we did not analyze the economic effect as conducted by Inage et al. [27]. This will be addressed in future studies when more robust findings are achieved with use of our developed model. It can be expected that further reduction in false positive rates will make a significant contribution to reducing ICA procedures in the clinical practice. 

## 5. Conclusions

In conclusion, we demonstrated significant improvements in the diagnostic performance of CCTA with the use of advanced DL approach, the finetuned Real-ESRGAN model for suppressing the blooming artifacts associated with severe calcification in the coronary arteries, with increased specificity and PPV than the previous studies. The Real-ESRGAN-Median images led to the higher diagnostic value with the specificity and PPV reaching 80% and 62%, respectively, at the RCA, although low to moderate diagnostic values at LAD and LCx. The false positive rates were significantly reduced when assessing calcified plaques at the three main coronary arteries, however, false negative findings should not be ignored. Further studies in large datasets are needed to validate our findings with potential clinical impact on economic benefits and patient management.

## Figures and Tables

**Figure 1 jpm-12-01354-f001:**
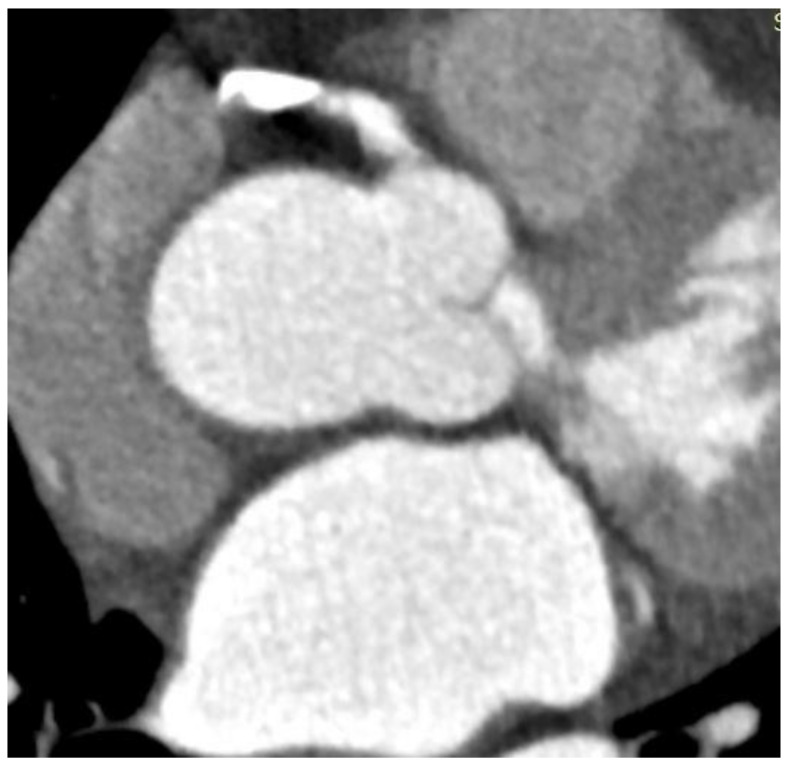
Heavy calcification in the proximal segment of right coronary artery prevents accurate assessment of coronary lumen and degree of stenosis due to blooming artifact.

**Figure 3 jpm-12-01354-f003:**
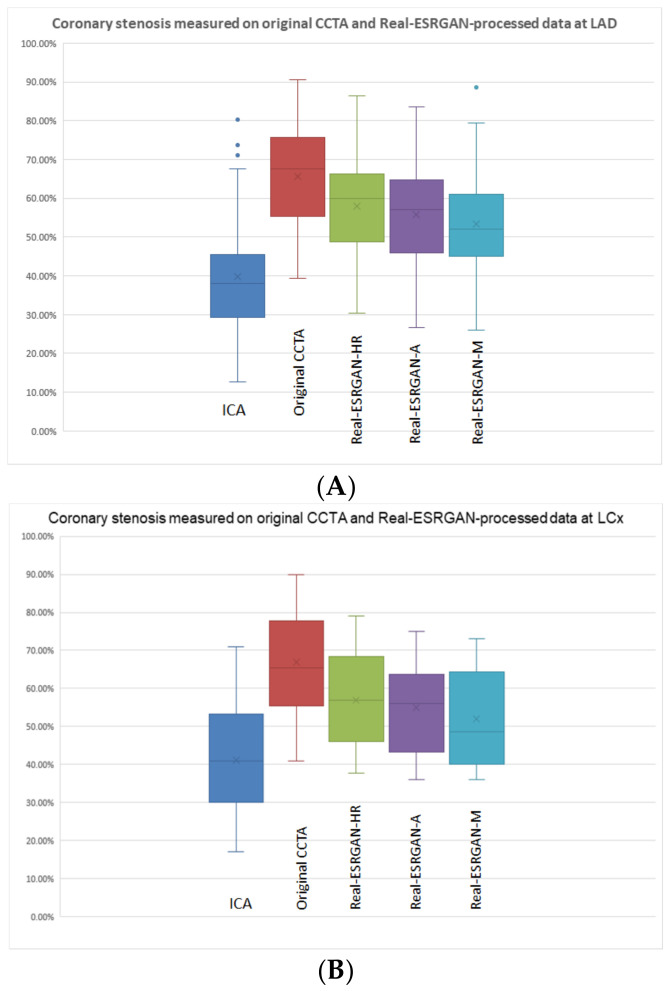
Boxplot showing the comparison of coronary stenosis measurements at LAD (**A**), LCx (**B**) and RCA (**C**) on original CCTA and Real-ESRGAN-processed images with ICA as the reference. Both original CCTA and Real-ESRGAN-processed images significantly overestimated the degree of stenosis at these three coronary arteries, however, the Real-ESRGAN-M images showed the best improvement compared to the original and Real-ESRGAN-HR and Real-ESRGAN-A images. The blue dots in (**A**,**C**) indicate the outliers, as some cases had coronary stenosis more than 70%, which is outside the average range distribution of coronary stenosis in this group. A—average; CCTA—coronary computed tomography angiography; ESRGAN—enhanced super-resolution generative adversarial network; HR—high resolution; ICA—invasive coronary angiography; LAD—left anterior descending; LCx—left circumflex; M—median; RCA—right coronary artery.

**Figure 4 jpm-12-01354-f004:**
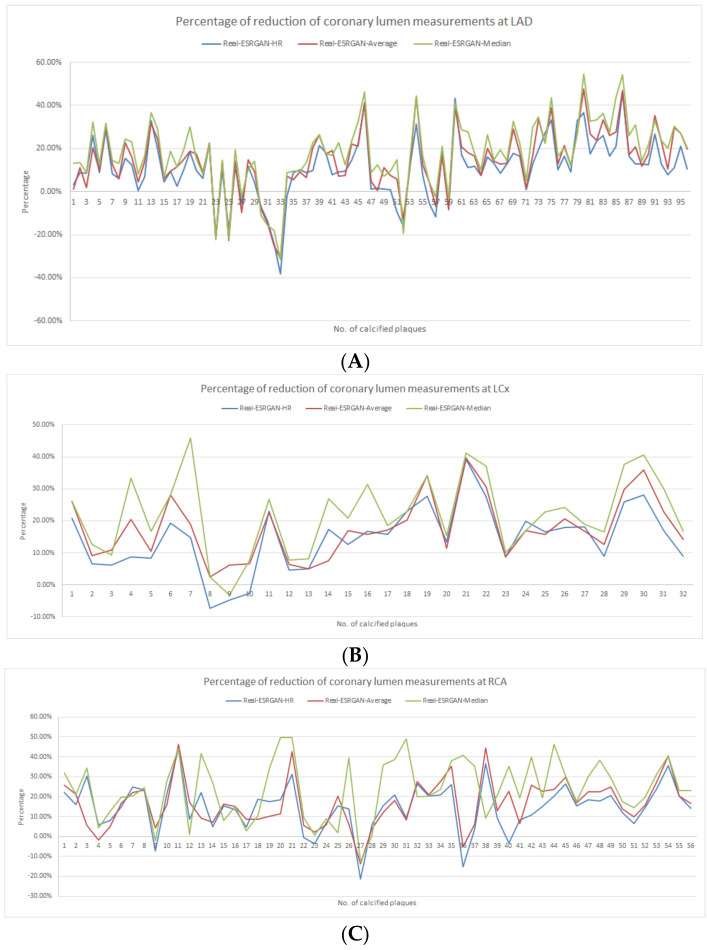
Graphs showing the percentage reduction when assessing coronary stenosis at LAD (**A**), LCx (**B**) and RCA (**C**) with use of Real-ESRGAN-processed images when compared to original CCTA. Real-ESRGAN-Median resulted in a higher reduction than the Real-ESRGAN-HR and Real-ESRGAN-Average. CCTA—coronary computed tomography angiography; ESRGAN—enhanced super-resolution generative adversarial network; HR—high resolution; LAD—left anterior descending; LCx—left circumflex; No.—number; RCA—right coronary artery.

**Figure 5 jpm-12-01354-f005:**
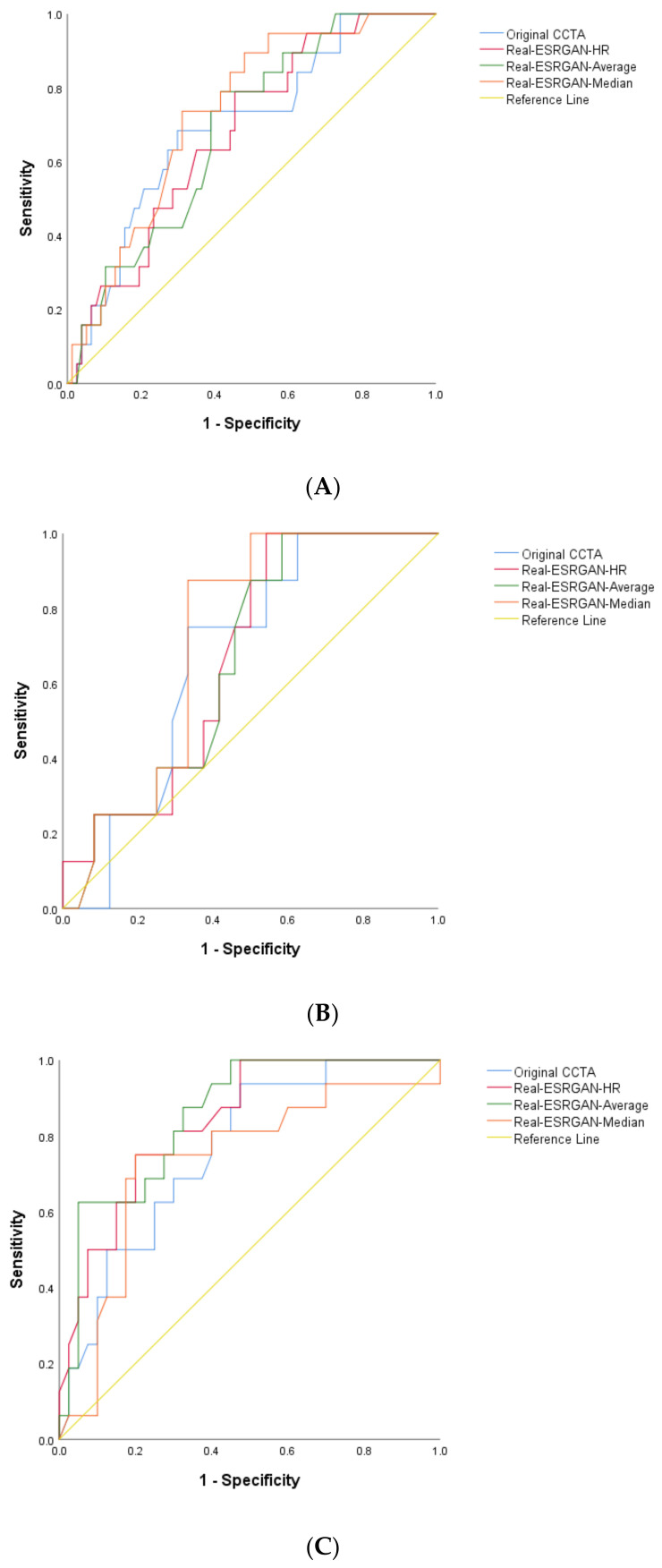
AUC of ROC analysis between original CCTA and Real-ESRGAN-processed images in the diagnosis of calcified plaques at LAD (**A**), LCx (**B**) and RCA (**C**). The AUC was the highest at the RCA level achieving 0.84 and 0.85 with Real-ESRGAN-HR and Real-ESRGAN-Average, respectively, but slightly lower for Real-ESRGAN-Median (0.73) due to false negative rates. AUC—area under curve; CCTA—coronary computed tomography angiography; ESRGAN—enhanced super-resolution generative adversarial network; HR—high resolution; LAD—left anterior descending; LCx—left circumflex; RCA—right coronary artery; ROC—receiver operating characteristic.

**Figure 6 jpm-12-01354-f006:**
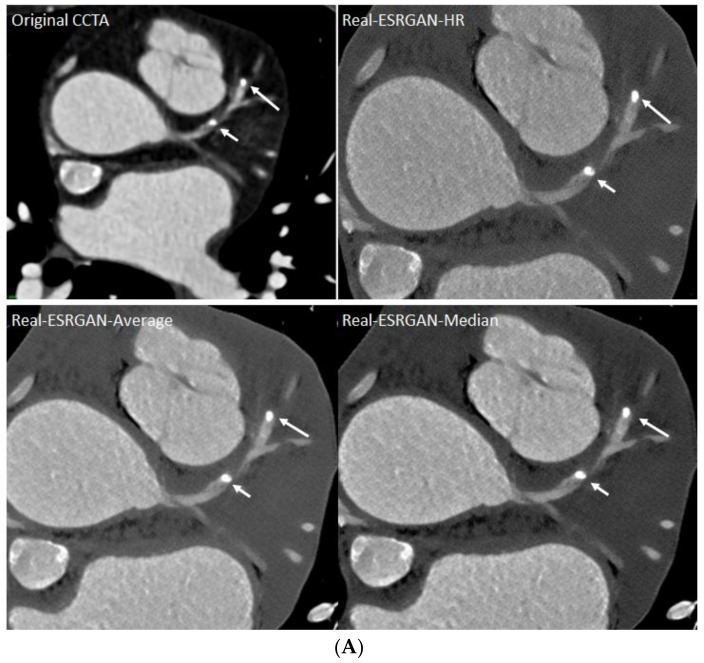
Multiple calcified plaques at the left anterior descending (LAD) in a 72-year-old female with coronary artery disease. The proximal calcified plaque resulted in significant stenosis as observed on original CCTA and Real-ESRGAN-processed images with stenosis measured as 80%, 78%, 72% and 70% corresponding to original CCTA, Real-ESRGAN-HR, Real-ESRGAN-Average and Real-ESRGAN-Median images (short arrows in **A**), respectively. ICA (short arrow in **B**) confirms the stenosis of 75%. The distal calcified plaque at LAD resulted in 70%, 50% and 51% stenosis on original CCTA, Real-ESRGAN-HR and Real-ESRGAN-Average images but was measured at 45% on Real-ESRGAN-Median image (long arrows in **A**). This was confirmed as 37% stenosis on ICA (long arrow in **B**). CCTA—coronary computed tomography angiography; ESRGAN—enhanced super-resolution generative adversarial network; HR—high resolution; ICA—invasive coronary angiography.

**Figure 7 jpm-12-01354-f007:**
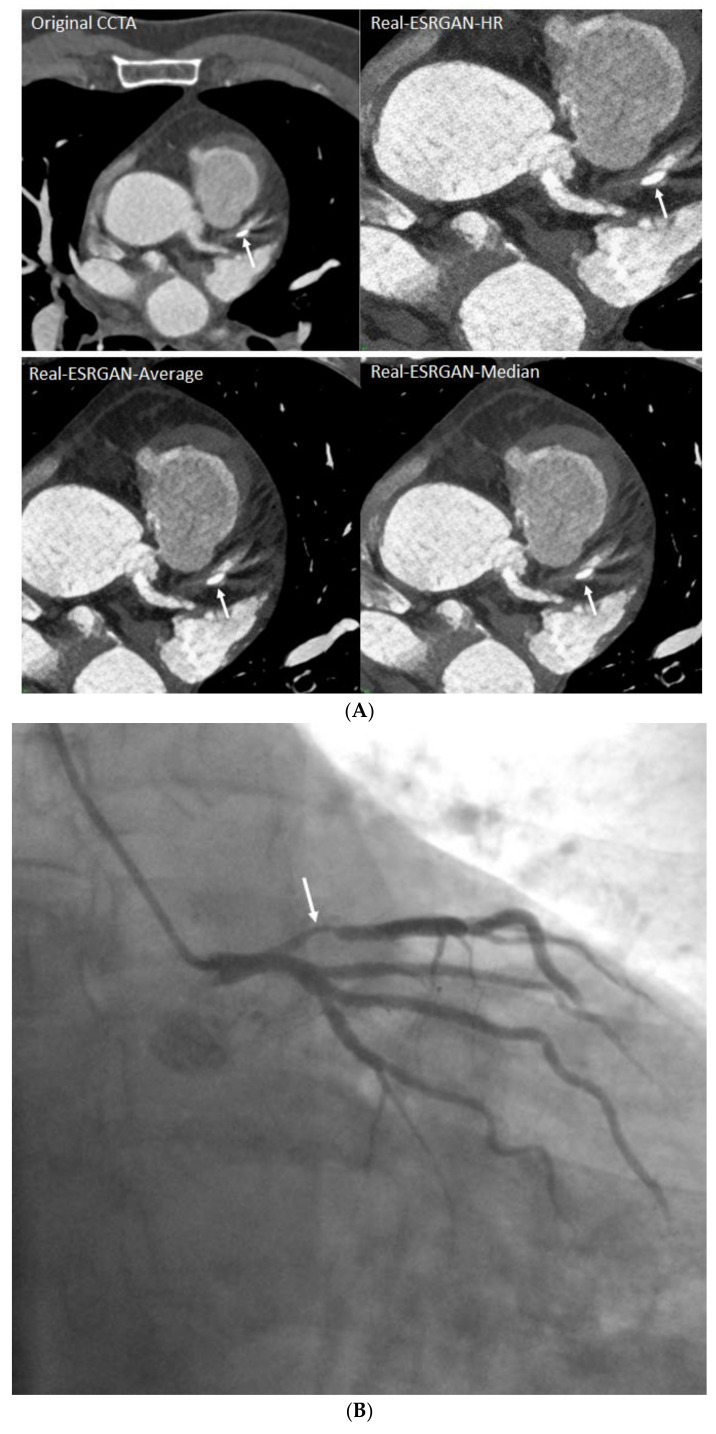
A calcified plaque at the proximal segment of left anterior descending in a 70-year-old man with coronary artery disease. The calcified plaque was measured 60%, 51%, 47% and 48% on original CCTA, Real-ESRGAN-HR, Real-ESRGAN-Average and Real-ESRGAN-Median images (arrows in **A**), and this was confirmed as 60% on invasive coronary angiography (arrow in **B**). The Real-ESRGAN-Average and Real-ESRGAN-Median images resulted in false negative finding. CCTA—coronary computed tomography angiography; ESRGAN—enhanced super-resolution generative adversarial network; HR—high resolution.

**Table 1 jpm-12-01354-t001:** Diagnostic value of original CCTA and Real-ESRGAN-processed images for assessment of calcified plaques on per-vessel assessment with ICA as the reference.

Coronary Arteries/No. Plaques	TP	FP	TN	FN	Sensitivity (%)	Specificity (%)	PPV (%)	NPV (%)	PLR	NLR	AUC
LAD											
Original CCTA	19	60	17	0	100 (82.3, 100)	22.1 (13.4, 32.9)	24.1 (21.9, 26.3)	100	1.28 (1.13, 1.44)	0.00	0.69 (0.57, 0.82)
Real-ESRGAN-HR	18	51	26	1	94.7 (73.9, 99.8)	33.8 (23.4, 45.4)	26.1 (22.5, 29.9)	96.3 (78.9, 99.4)	1.43 (1.18, 1.73)	0.16 (0.02, 1.08)	0.68 (0.56, 0.80)
Real-ESRGAN-Average	17	47	30	2	89.5 (66.8, 98.7)	38.9 (28.0, 50.7)	26.6 (22.2, 31.4)	93.7 (79.7, 98.3)	1.47 (1.16, 1.86)	0.27 (0.07, 1.03)	0.69 (0.57, 0.80)
Real-ESRGAN-Median	17	37	40	2	89.5 (66.9, 98.7)	51.9 (40.3, 63.5)	31.5 (25.8, 37.8)	95.2 (84.1, 98.7)	1.86 (1.41, 2.46)	0.20 (0.05, 0.77)	0.73 (0.62, 0.85)
**LCx**											
Original CCTA	8	21	3	0	100 (63.1, 100)	12.5 (2.6, 32.4)	27.6 (24.7, 30.7)	100	1.14 (0.98, 1.33)	0.00	0.67 (0.48, 0.86)
Real-ESRGAN-HR	8	13	11	0	100 (63.1, 100)	45.8 (25.6, 67.2)	38.1 (29.9, 47.1)	100	1.85 (1.28, 2.67)	0.00	0.67 (0.48, 0.86)
Real-ESRGAN-Average	7	13	11	1	87.5 (47.3 99.7)	45.8 (25.5, 67.2)	35.0 (25.5, 45.8)	91.7 (62.6, 98.6)	1.62 (1.03, 2.54)	0.27 (0.04, 1.79)	0.66 (0.47, 0.85)
Real-ESRGAN-Median	7	9	15	1	87.5 (47.3, 99.7)	62.5 (40.6, 81.2)	43.8 (30.3, 58.1)	93.7 (70.0, 98.9)	2.33 (1.31, 4.16)	0.20 (0.03, 1.28)	0.72 (0.55, 0.89)
**RCA**											
Original CCTA	16	33	7	0	100 (79.4, 100)	17.5 (7.3, 32.8)	32.7 (29.6, 35.9)	100	1.21 (1.05, 1.39)	0.00	0.76 (0.64, 0.89)
Real-ESRGAN-HR	16	20	20	0	100 (79.4, 100)	50.0 (33.8, 66.2)	44.4 (36.9, 52.2)	100	2.00 (1.47, 2.73)	0.00	0.84 (0.73, 0.94)
Real-ESRGAN-Average	15	16	24	1	93.7 (69.8, 99.8)	60.0 (43.3, 75.1)	48.4 (38.6, 58.3)	96.0 (77.9, 99.4)	2.34 (1.57, 3.50)	0.10 (0.02, 0.71)	0.85 (0.75, 0.95)
Real-ESRGAN-Median	13	8	32	3	81.3 (54.4, 95.9)	80.0 (64.4, 90.9)	61.9 (45.6, 75.9)	91.4 (79.2, 96.7)	4.06 (2.09, 7.88)	0.23 (0.08, 0.66)	0.73 (0.58, 0.89)

Numbers in brackets indicate 95% confidence interval. AUC—area under the receiver operating characteristic curve; CCTA—coronary computed tomography angiography; ESRGAN—enhanced super-resolution generative adversarial network; FN—false negative; FP—false positive; HR—high resolution; ICA—invasive coronary angiography, LAD—left anterior descending artery; LCx—left circumflex artery; NLR—negative likelihood ratio; No.—number; NPV—negative predictive value; PLR—positive likelihood ratio; PPV—positive predictive value; RCA—right coronary artery; TN—true negative; TP—true positive.

## Data Availability

The datasets used in this study are not publicly available due to strict requirements set out by authorized investigators.

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
