# Peer review of "Finetuned Super-Resolution Generative Adversarial Network (Artificial Intelligence) Model for Calcium Deblooming in Coronary Computed Tomography Angiography"

_jpm, 2022, doi:10.3390/jpm12091354_

Round 1

Reviewer 1 Report

Paper is important for health informatics. Following revisions are to be incorporated before publication-

(1)   How data sets are established for doing this research work?

(2)   Which files of datasets are using in this paper? Mention these websites.

(3)   What are the strong features of this research work? Author must explain.

(4)   Author should add the motivations, problem, and solution statement in the abstract.

(5)   How the parameters for simulations are selected?

(6)   How the performance of proposed technique is better than existing techniques.

(7)   All tables and figures should be explained clearly.

(8)   The English and typo errors of the paper should be checked in the presence of native English speaker.

(9)   All equations should be clearly explained with explanation on all associated variables.

(10) Author should add one section “Related Work” in the paper.

(11) The methodology of the paper should be clearly explained with appropriate flow charts.

(12) Highlight the more applications of the proposed technique.

(13) What are motivations behind this research work?

(14) Add more explanation on obtained results with critical analysis.

(15) Author must explain pros and cons of the work.

(16) What are the major issues in the AI?

(17) How over fitting is minimized in the proposed work?

(18) Author must cite suggested papers for enhancing the quality of the paper. These are based on realted techniques and confusion matrices-

(a)    Multi-Feature Fusion Method for Identifying Carotid Artery Vulnerable Plaque

(b)   An efficient AR modelling-based electrocardiogram signal analysis for health informatics

(c)    Prediction of atherosclerosis pathology in retinal fundal images with machine learning approaches

(d)   3D Coronary Artery Reconstruction by 2D Motion Compensation Based on Mutual Information

(e)    Robust retinal blood vessel segmentation using convolutional neural network and support vector machine

(f)    Real-time estimation of hospital discharge using fuzzy radial basis function network and electronic health record data

(g)   An efficient ALO-based ensemble classification algorithm for medical big data processing

(h)   Non-invasive assessment of fractional flow reserve using computational fluid dynamics modelling from coronary angiography images

(i)    Bone metastatic tumourminimisation due to thermal cementoplasty effect, clinical and computational methodologies

(j)    Multiscale Graph Cuts Based Method for Coronary Artery Segmentation in Angiograms

(k)   Bio-medical analysis of breast cancer risk detection based on deep neural network

(l)    Changes in scale-invariance property of electrocardiogram as a predictor of hypertension

(m)  Peak alpha neurofeedback training on cognitive performance in elderly subjects

(n)   Modified model for cancer treatment

(o)   Assessment of qualitative and quantitative features in coronary artery MRA

(p)   A frugal and innovative telemedicine approach for rural India – automated doctor machine

(q)   Study of murmurs and their impact on the heart variability

(r)    Analysis of salivary components as non-invasive biomarkers for monitoring chronic kidney disease

(s)    Coronary three-vessel disease with occlusion of the right coronary artery: What are the most important factors that determine the right territory perfusion?

(t)    An improved graph matching algorithm for the spatio-temporal matching of a coronary artery 3D tree sequence

Author Response

Reviewer 1

Paper is important for health informatics. Following revisions are to be incorporated before publication-

Response: Thank you for your comment.

(1) How data sets are established for doing this research work?

Response: Thank you for your comment. Datasets were used for our real-enhanced super-resolution generative adversarial network (Real-ESRGAN) model development at three different stages. These data sets were obtained from three different sources and the associated details have been given in the manuscript as follows.

Section 2.1.: “Although the same publicly available datasets used for training the original ESRGAN, i.e. DIV2K (https://data.vision.ee.ethz.ch/cvl/DIV2K/, accessed on 1 February 2022), Flickr2K (http://cv.snu.ac.kr/research/EDSR/Flickr2K.tar, accessed on 1 February 2022) and OutdoorSceneTraining (OST) (http://mmlab.ie.cuhk.edu.hk/projects/SFTGAN/, accessed on 1 February 2022) with a total of 13,774 non-medical images were used for training the Real-ESRGAN model with 400,000 iterations, these images were sharpened before using them as ground-truth images for the training [13].”

Section 2.1.: “For finetuning Wang et al.’s Real-ESRGAN model to achieve better performance on calcium deblooming [13], 32 deidentified CCTA datasets acquired by a 640-slice CT scanner (Toshiba Aquilion ONE, Toshiba, Otawara, Japan) in 2015 with a reconstruction slice thickness of 0.5 mm and interval of 0.25 mm in Digital Imaging and Communications in Medicine (DICOM) format of patients with heavy calcification in the coronary arteries were collected.”

Section 2.2.: “The finetuned Real-ESRGAN model was used to postprocess 50 CCTA datasets not in-volved in the finetuning process, i.e. to increase the image spatial resolution from 512 x 512 pixels (original resolution) to 2048 x 2048 pixels (high-resolution) for calcium deblooming. Approximately 10 minutes was needed to postprocess each dataset (hundreds of images) with the use of the Kaggle platform. These 50 CCTA datasets with paired reference images (ICA datasets) were those used in our previous study for evaluating the performance of the ESRGAN model on blooming artifact suppression. Our previous approach to evaluate the ESRGAN model was also adopted in this study. Details of these datasets and evaluation strategy were available from our previous article.”

We hope our response provides an adequate clarification for addressing the comment, “How data sets are established for doing this research work?”

(2) Which files of datasets are using in this paper? Mention these websites.

Response: Thank you for your comment. All files of the datasets mentioned in the manuscript were used and the URLs of the publicly available datasets have been provided within the paper while the others are not publicly available and we cannot provide the URLs. The contents given in the paper which are related to this comment are shown below.

Section 2.1.: “Although the same publicly available datasets used for training the original ESRGAN, i.e. DIV2K (https://data.vision.ee.ethz.ch/cvl/DIV2K/, accessed on 1 February 2022), Flickr2K (http://cv.snu.ac.kr/research/EDSR/Flickr2K.tar, accessed on 1 February 2022) and OutdoorSceneTraining (OST) (http://mmlab.ie.cuhk.edu.hk/projects/SFTGAN/, accessed on 1 February 2022) with a total of 13,774 non-medical images were used for training the Real-ESRGAN model with 400,000 iterations, these images were sharpened before using them as ground-truth images for the training [13].”

Section 2.1.: “For finetuning Wang et al.’s Real-ESRGAN model to achieve better performance on calcium deblooming [13], 32 deidentified CCTA datasets acquired by a 640-slice CT scanner (Toshiba Aquilion ONE, Toshiba, Otawara, Japan) in 2015 with a reconstruction slice thickness of 0.5mm and interval of 0.25mm in Digital Imaging and Communications in Medicine (DICOM) format of patients with heavy calcification in the coronary arteries were collected...The 32 datasets consisted of 16,904 images with an equal size of 512 x 512 pixels. All datasets were used to train both generator and discriminator of the Real-ESRGAN model with 40,000 iterations through the free Kaggle platform (Google LLC, Mountain View, CA, USA) with one NVidia K80 graphics processing unit (Santa Clara, CA, USA).”

Section 2.2.: “The finetuned Real-ESRGAN model was used to postprocess 50 CCTA datasets not in-volved in the finetuning process, i.e. to increase the image spatial resolution from 512 x 512 pixels (original resolution) to 2048 x 2048 pixels (high-resolution) for calcium deblooming. Approximately 10 minutes was needed to postprocess each dataset (hundreds of images) with the use of the Kaggle platform. These 50 CCTA datasets with paired reference images (ICA datasets) were those used in our previous study for evaluating the performance of the ESRGAN model on blooming artifact suppression. Our previous approach to evaluate the ESRGAN model was also adopted in this study. Details of these datasets and evaluation strategy were available from our previous article.”

“Data Availability Statement: The datasets used in this study are not publicly available due to strict requirements set out by authorized investigators.”

We hope our response provides an adequate clarification for addressing the comment, “Which files of datasets are using in this paper? Mention these websites.”

(3) What are the strong features of this research work? Author must explain.

Response: Thank you for your comment. The strong features of our research work have been discussed and explained in Section 4. The details are as follows.

“This study further advances our recent report of using the finetuned DL model, Real-ESRGAN to postprocess the original CCTA images with results showing significant improvements over previous studies [3-5]. Based on analysis of the same dataset, our results showed that the specificity and PPV were further increased by up to 25% and 15%, respectively compared to our recent results [3], indicating that the finetuned Real-ESRGAN model allows for further improvement in assessing calcified coronary plaques. This has significant clinical impact as the number of false positive rates were reduced, thus reducing the unnecessary downstream testing such as avoiding ICA procedures when diagnosing calcified coronary plaques.”

“Inage et al. [27] in their recent study applied the cycle GAN-based lumen extraction model to CCTA images in 99 patients involving 891 segments with severe calcification in the coronary arteries. Diagnostic value of assessing coronary stenosis by the original CCTA and cycle GAN-processed images were compared with ICA as the reference method. In addition to assessing the performance of original CCTA and cycle GAN-processed images in all 891 segments, authors focused on the analysis of 228 segments which were not assessable on the original CCTA images due to severe calcification. Their results showed similar specificity and PPV between the original CCTA and cycle GAN-processed images (75.1% and 40.9% vs. 77.3% and 43.4%) among assessment of all coronary segments, with AUC significantly higher in the cycle GAN group than the original CCTA group (0.77 vs. 0.75, p=0.03). For the non-assessable 228 segments, the cycle GAN model significantly improved the specificity and accuracy compared to the original CCTA (10.9% and 42.5% vs. 0% and 35.5%, p<0.001), along with significantly higher AUC (0.59 vs. 0.50, p<0.001). Similar to assessment of all segments, the PPV was similar between these two groups (35.5% and 38.2% for the original CCTA and cycle GAN). Authors claimed that the use of cycle GAN model could avoid four out of 99 ICA examinations based on their study, with estimated 747 ICA procedures to be avoided per year. Despite promising results achieved within this study, the PPV was low (<45%) with moderate specificity and very low specificity in all segments and non-assessable segments groups. In contrast, our findings showed much better results than those from Inage et al.’s study [27]. Both specificity and PPV were increased significantly with the use of finetuned Real-ESRGAN model, achieving 80% and 61% at the RCA level, although still low at LAD and LCx levels. This represents so far the most promising outcomes of diagnosing calcified plaques.”

We hope our response provides an adequate clarification for addressing the comment, “What are the strong features of this research work? Author must explain.”

(4) Author should add the motivations, problem, and solution statement in the abstract.

Response: Thank you for your comment. Ideas related to motivations, problem and solution have been embedded in the first two sentences of the Abstract, “The purpose of this study was to finetune a deep learning model, real-enhanced super-resolution generative adversarial network (Real-ESRGAN) and investigate its diagnostic value in calcified coronary plaques with the aim of suppressing blooming artifacts for further improvement of coronary lumen assessment. We finetuned the Real-ESRGAN model and applied it to 50 patients with 184 calcified plaques detected at three main coronary arteries (left anterior descending [LAD], left circumflex [LCx] and right coronary artery [RCA]).” Our further clarification is given below.

Motivation: “…suppressing blooming artifacts for further improvement of coronary lumen assessment”

Problem: “…blooming artifacts…”

Solution: “…finetuned the Real-ESRGAN model and applied it to 50 patients with 184 calcified plaques detected at three main coronary arteries (left anterior descending [LAD], left circumflex [LCx] and right coronary artery [RCA]).”

Due to the Journal of Personalized Medicine Instructions for Authors (https://www.mdpi.com/journal/jpm/instructions), “the abstract should be a total of about 200 words maximum.”, apparently, it is not practical for providing statements of these in the Abstract and it is expected that motivations, problem and solution points would be sufficient. We hope our response provides an adequate clarification for addressing the comment, “Author should add the motivations, problem, and solution statement in the abstract.”

(5) How the parameters for simulations are selected?

Response: Thank you for your comment. However, our study appears not related to simulations. The words, “simulation” and “simulate” do not appear within our manuscript. Please clarify your comment if necessary.

(6) How the performance of proposed technique is better than existing techniques.

Response: Thank you for your comment. Details of “How the performance of proposed technique is better than existing techniques.” have been given in Section 4 as follows.

“This study further advances our recent report of using the finetuned DL model, Real-ESRGAN to postprocess the original CCTA images with results showing significant improvements over previous studies [3-5]. Based on analysis of the same dataset, our results showed that the specificity and PPV were further increased by up to 25% and 15%, respectively compared to our recent results [3], indicating that the finetuned Real-ESRGAN model allows for further improvement in assessing calcified coronary plaques. This has significant clinical impact as the number of false positive rates were reduced, thus reducing the unnecessary downstream testing such as avoiding ICA procedures when diagnosing calcified coronary plaques.”

“Inage et al. [27] in their recent study applied the cycle GAN-based lumen extraction model to CCTA images in 99 patients involving 891 segments with severe calcification in the coronary arteries. Diagnostic value of assessing coronary stenosis by the original CCTA and cycle GAN-processed images were compared with ICA as the reference method. In addition to assessing the performance of original CCTA and cycle GAN-processed images in all 891 segments, authors focused on the analysis of 228 segments which were not assessable on the original CCTA images due to severe calcification. Their results showed similar specificity and PPV between the original CCTA and cycle GAN-processed images (75.1% and 40.9% vs. 77.3% and 43.4%) among assessment of all coronary segments, with AUC significantly higher in the cycle GAN group than the original CCTA group (0.77 vs. 0.75, p=0.03). For the non-assessable 228 segments, the cycle GAN model significantly improved the specificity and accuracy compared to the original CCTA (10.9% and 42.5% vs. 0% and 35.5%, p<0.001), along with significantly higher AUC (0.59 vs. 0.50, p<0.001). Similar to assessment of all segments, the PPV was similar between these two groups (35.5% and 38.2% for the original CCTA and cycle GAN). Authors claimed that the use of cycle GAN model could avoid four out of 99 ICA examinations based on their study, with estimated 747 ICA procedures to be avoided per year. Despite promising results achieved within this study, the PPV was low (<45%) with moderate specificity and very low specificity in all segments and non-assessable segments groups. In contrast, our findings showed much better results than those from Inage et al.’s study [27]. Both specificity and PPV were increased significantly with the use of finetuned Real-ESRGAN model, achieving 80% and 61% at the RCA level, although still low at LAD and LCx levels. This represents so far the most promising outcomes of diagnosing calcified plaques.”

We hope our response provides an adequate clarification for addressing the comment, “How the performance of proposed technique is better than existing techniques.”

(7) All tables and figures should be explained clearly.

Response: Thank you for your comment. Clear explanations of all tables and figures have been provided in Section 3 as follows.

“There were a total of 184 calcified plaques that were assessed in this study with the same plaque distribution at these three main coronary arteries as reported in our recent study [3]. Compared to the measurements on ICA, original CCTA and Real-ESRGAN-processed images overestimated the degree of coronary stenosis resulting in significant differences in coronary lumen measurements (p<0.01), as shown in Figure 3.”

“Figure 3. Boxplot showing the comparison of coronary stenosis measurements at LAD (A), LCx (B) and RCA (C) on original CCTA and Real-ESRGAN-processed images with ICA as the reference. Both original CCTA and Real-ESRGAN-processed images significantly overestimated the degree of stenosis at these three coronary arteries, however, the Real-ESRGAN-M images showed the best improvement compared to the original and Real-ESRGAN-HR and Real-ESRGAN-A images. The blue dots in A and C indicate the outliners as some cases had coronary stenosis more than 70% which is outside the average range distribution of coronary stenosis in this group. A-average; CCTA-coronary computed tomography angiography; ESRGAN-enhanced super-resolution generative adversarial network; HR-high-resolution; ICA-invasive coronary angiography; LAD-left anterior descending; LCx-left circumflex; M-median; RCA-right coronary artery.”

“Of these Real-ESRGAN-processed images, the Real-ESRGAN-Median images showed the most significant improvements in the degree of reducing blooming artifacts, with the mean value and SD being 10.99 ± 13.94%, 14.42 ± 14.78% and 18.06 ± 15.74% at LAD; 14.57 ± 10.13%, 17.53 ± 9.64% and 22.02 ± 12.02% at LCx; 14.74 ± 11.90%, 16.63 ± 12.01% and 23.81 ± 14.96% at RCA, corresponding to Real-ESRGAN-HR, Real-ESRGAN-Average and Real-ESRGAN-Median images, respectively. Figure 4 shows the percentage of reduction of coronary lumen measurements at the three main coronary arteries as assessed by these three Real-ESRGAN-processed images when compared to those from the original CCTA images. Although the Real-ESRGAN-Median images led to the highest degree of reduction in most of the plaque assessments (>90%), indicating its significant impact on suppressing the blooming artifact, increased overestimation of the coronary lumen (by 11-32%) was observed in 7 plaques at LAD (plaque numbers 23, 25, 30-33 and 52) when compared to the original CCTA and other Real-ESRGAN-processed images. In contrast, this phenomenon was not observed at LCx and only in two plaques (plaques numbers 27 and 36) at RCA (Figure 4).”

“Figure 4. Graphs showing the percentage reduction when assessing coronary stenosis at LAD (A), LCx (B) and RCA (C) with use of Real-ESRGAN-processed images when compared to original CCTA. Real-ESRGAN-Median resulted in the highest reduction than the Real-ESRGAN-HR and Real-ESRGAN-Average. CCTA-coronary computed tomography angiography; ESRGAN-enhanced super-resolution generative adversarial network; HR-high-resolution; LAD-left anterior descending; LCx-left circumflex; No.-number; RCA-right coronary artery.”

“The number of false positive rates was found highest in the original CCTA images resulting in the lowest specificity and PPV at all three coronary arteries as shown in Table 1. The number of false positive rates was reduced when applying the Re-al-ESRGAN model to postprocess the original CCTA images, with Re-al-ESRGAN-Median images showing the significant impact on reducing the false positive rates. The specificity and PPV were significantly improved with Re-al-ESRGAN-Median images compared to original CCTA, Real-ESRGAN-HR and Re-al-ESRGAN-Average images at all three coronary arteries (Table 1). With use of Re-al-ESRGAN-Median images, the specificity and PPV achieved 80% and 61.9% at RCA, 52-62% and 32-44% at LAD and LCx, respectively, although false negative cases were found in the Real-ESRGAN-processed images which decreased the sensitivity to some extent. The area under curve (AUC) of ROC analysis was higher in Re-al-ESRGAN-processed images than that in original CCTA as shown in Figure 5. The highest AUC was found in Real-ESRGAN-processed images at RCA level (Table 1).”

“Figure 5. AUC of ROC analysis between original CCTA and Real-ESRGAN-processed images in the diagnosis of calcified plaques at LAD, LCx and RCA (A-C). The AUC was the highest at the RCA level achieving 0.84 and 0.85 with Real-ESRGAN-HR and Real-ESRGAN-Average respectively, but slightly lower for Real-ESRGAN-Median (0.73) due to false negative rates. AUC-area under curve; CCTA-coronary computed tomography angiography; ESRGAN-enhanced super-resolution generative adversarial network; HR-high-resolution; LAD-left anterior descending; LCx-left circum-flex; RCA-right coronary artery; ROC-receiver operating characteristic.”

“Figure 6 is an example of multiple calcified plaques at LAD with improved visualization of coronary lumen observed in Real-ESRGAN-processed images, while Figure 7 is another example showing the calcified plaques at LAD with Real-ESRGAN-processed images resulting in false negative finding when compared to original CCTA and ICA images.”

“Figure 6. Multiple calcified plaques at the left anterior descending (LAD) in a 72-year-old female with coronary artery disease. The proximal calcified plaque resulted in significant stenosis as ob-served on original CCTA and Real-ESRGAN-processed images with stenosis measured as 80%, 78%, 72% and 70% corresponding to original CCTA, Real-ESRGAN-HR, Real-ESRGAN-Average and Real-ESRGAN-Median images (short arrows in A), respectively. ICA (short arrow in B) con-firms the stenosis of 75%. The distal calcified plaque at LAD resulted in 70%, 50% and 51% stenosis on original CCTA, Real-ESRGAN-HR and Real-ESRGAN-Average images, but was measured 45% on Real-ESRGAN-Median image (long arrows in A). This was confirmed as 37% stenosis on ICA (long arrow in B). CCTA-coronary computed tomography angiography; ESRGAN-enhanced super-resolution generative adversarial network; HR-high-resolution; ICA-invasive coronary angiography.”

“Figure 7. A calcified plaque at the proximal segment of left anterior descending in a 70-year-old man with coronary artery disease. The calcified plaque was measured 60%, 51%, 47% and 48% on original CCTA, Real-ESRGAN-HR, Real-ESRGAN-Average and Real-ESRGAN-Median images (arrows in A), and this was confirmed as 60% on invasive coronary angiography (arrow in B). The Real-ESRGAN-Average and Real-ESRGAN-Median images resulted in false negative finding. CCTA-coronary computed tomography angiography; ESRGAN-enhanced super-resolution generative adversarial network; HR-high-resolution.”

“Table 1. Diagnostic value of original CCTA and Real-ESRGAN-processed images for assessment of calcified plaques on per-vessel assessment with ICA as the reference … Numbers in brackets indicate 95% confidence interval. AUC-area under the receiver operating characteristic curve; CCTA-coronary computed tomography angiography; ESRGAN- enhanced super-resolution generative adversarial network; FN-false negative; FP-false positive; HR-high-resolution; ICA-invasive coronary angiography, LAD-left anterior descending artery; LCx- left circumflex artery; NLR-negative likelihood ratio; No.-number; NPV-negative predictive value; PLR-positive likelihood ratio; PPV-positive predictive value; RCA-right coronary artery; TN-true negative; TP-true positive.”

We hope our response provides an adequate clarification for addressing the comment, “All tables and figures should be explained clearly.”

(8) The English and typo errors of the paper should be checked in the presence of native English speaker.

Response: Thank you for your comment. The English and typo errors of the paper have been checked and necessary changes have been made (highlighted in the revised manuscript).

(9) All equations should be clearly explained with explanation on all associated variables.

Response: Thank you for your comment. Only one formula was given in our manuscript (Section 2.2.). Clear explanations of this formula and all associated variables have been given in the same section, i.e. Section 2.2. as follows.

“Measurements of minimal lumen diameter (MLD) at each calcified plaque lesion of three main coronary arteries, left anterior descending (LAD), left circumflex (LCx) and right coronary artery (RCA) for the 200 datasets (50 original CCTA, 50 Real-ESRGAN-HR, 50 Real-ESRGAN-Average and 50 Real-ESRGAN-Median datasets) by a single researcher (with experience more than 20 years in CCTA image interpretation) for three times per lesion with average value taking as the final. The MLD was measured at the narrowest part of each coronary lumen (the most extensively calcified area) to determine the degree of stenosis on the original CCTA and Real-ESRGAN-processed images with measurements on ICA as the reference to calculate the diagnostic value. Determination of blooming artifact reduction by using formula (1) below.

[(MLDReal-ESRGAN-Processed Image-MLDOriginal CCTA Image)/MLDOriginal CCTA Image] x 100% ”

We hope our response provides an adequate clarification for addressing the comment, “All equations should be clearly explained with explanation on all associated variables.”

(10) Author should add one section “Related Work” in the paper.

Response: Thank you for your comment. As per the Journal of Personalized Medicine Instructions for Authors (https://www.mdpi.com/journal/jpm/instructions) and manuscript template (https://www.mdpi.com/files/word-templates/jpm-template.dot), the “Related Work” section is not required. However, contents about “Related Work” have been given in Sections 1 and 4 as follows.

Section 1: “One of the main approaches for blooming artifact suppression is to improve the CCTA image spatial resolution. Various strategies for increasing the CCTA image spatial resolution to reduce the blooming artifact have been reported [3-6]. The latest strategy is to use artificial intelligence (AI) (specifically deep learning [DL]) including convolutional neural network (CNN)-based CT image reconstruction kernels such as Canon Medical Systems Advanced Intelligent Clear-IQ Engine (AiCE) and generative adversarial network (GAN) model for image postprocessing to achieve this goal [3,7].”

Section 1: “Our recent study has shown that enhanced super-resolution GAN (ESRGAN) was able to effectively suppress the CCTA blooming artifact, and improve the specificity and PPV by 10-40% for patients with heavy calcification in the coronary arteries [3]. Its performance was better than the Canon AiCE reconstruction kernel which could only increase the PPV by about 10% [3,7]. Despite of these promising results, AI inference was used in our previous study, i.e. no medical image was used to train the ESRGAN model for the calcium deblooming task [3,8,9]. Hence, one straightforward way to further improve the performance of the ESRGAN model for this task is to finetune the model with use of CCTA images [3,8-11]. The use of finetuning (a subset of transfer learning) has become popular in the medical imaging field because of limited availability of medical images for training a model from scratch, and time- and resource-efficient but still being able to achieve superior performance on similar tasks [12].”

Section 4: “This study further advances our recent report of using the finetuned DL model, Real-ESRGAN to postprocess the original CCTA images with results showing significant improvements over previous studies [3-5]. Based on analysis of the same dataset, our results showed that the specificity and PPV were further increased by up to 25% and 15%, respectively compared to our recent results [3], indicating that the finetuned Real-ESRGAN model allows for further improvement in assessing calcified coronary plaques. This has significant clinical impact as the number of false positive rates were reduced, thus reducing the unnecessary downstream testing such as avoiding ICA procedures when diagnosing calcified coronary plaques.”

Section 4: “Assessment of coronary artery disease, in particular coronary calcification and coronary plaques is a well-recognized issue in CCTA which has drawn increasing attention in recent years to tackle this challenging area. Although a number of strategies have been implemented with some promising results, use of AI algorithm to process original CCTA images represents the most promising strategy in the recent literature [15-23]. Studies have shown that Al algorithm allows for accurate and efficient quantification of coronary calcium scores and assessment of coronary stenosis when compared to the standard manual approach or semi-automatic method [15-19]. Al is also shown to achieve good accuracy in characterizing plaque morphology and differentiating plaque from no plaque, or calcified from non-calcified plaques [20-23]. However, very limited research has been conducted so far with use of AI in suppressing heavy calcification in the coronary arteries for reducing blooming artifacts to improve lumen assessment. Further, most of the previous studies used the traditional CNN model, which is inferior to the advanced GAN approach. Most applications using the GAN approach focus on CT denoising [24], coronary artery disease risk categorization by quantifying calcium scoring [25], and automated registration of positron emission tomography-CT angiography images in imaging coronary artery disease [26]. Our study has addressed this gap by using the latest GAN model with promising results achieved.”

Section 4: “Inage et al. [27] in their recent study applied the cycle GAN-based lumen extraction model to CCTA images in 99 patients involving 891 segments with severe calcification in the coronary arteries. Diagnostic value of assessing coronary stenosis by the original CCTA and cycle GAN-processed images were compared with ICA as the reference method. In addition to assessing the performance of original CCTA and cycle GAN-processed images in all 891 segments, authors focused on the analysis of 228 segments which were not assessable on the original CCTA images due to severe calcification. Their results showed similar specificity and PPV between the original CCTA and cycle GAN-processed images (75.1% and 40.9% vs. 77.3% and 43.4%) among assessment of all coronary segments, with AUC significantly higher in the cycle GAN group than the original CCTA group (0.77 vs. 0.75, p=0.03). For the non-assessable 228 segments, the cycle GAN model significantly improved the specificity and accuracy compared to the original CCTA (10.9% and 42.5% vs. 0% and 35.5%, p<0.001), along with significantly higher AUC (0.59 vs. 0.50, p<0.001). Similar to assessment of all segments, the PPV was similar between these two groups (35.5% and 38.2% for the original CCTA and cycle GAN). Authors claimed that the use of cycle GAN model could avoid four out of 99 ICA examinations based on their study, with estimated 747 ICA procedures to be avoided per year. Despite promising results achieved within this study, the PPV was low (<45%) with moderate specificity and very low specificity in all segments and non-assessable segments groups. In contrast, our findings showed much better results than those from Inage et al.’s study [27]. Both specificity and PPV were increased significantly with the use of finetuned Real-ESRGAN model, achieving 80% and 61% at the RCA level, although still low at LAD and LCx levels. This represents so far the most promising outcomes of diagnosing calcified plaques. Although we did not evaluate the economic benefits in our study, the improved PPV with reduced false positive rates will lead to avoidance of more unnecessary ICA examinations.”

Section 4: “With further reduction of false positive rates leading to improved specificity and PPV with the use of finetuned Real-ESRGAN model compared to our previous report [3], the negative effect is the slightly decreased sensitivity due to the false negative rates. Up to three false negative cases were noticed in the Real-ESRGAN-processed images at all of three coronary arteries with the highest number seen in the Real-ESRGAN-Average and Real-ESRGAN-Median groups (Table 1), but not in the original CCTA images. This issue could be attributed to two factors. Firstly, only 32 datasets consisted of 16,904 CCTA images were used to finetune the Real-ESRGAN model. According to Wang et al.’s [11] study about transfer learning of pre-trained GAN models, this arrangement should be sufficient. The unexpected false negative cases would be an indication of the Real-ESRGAN model not exposed to adequate variations of plaque characteristics, e.g. varied compositions with presence of mixed calcium and atheromatous plaques, etc. More cases with greater varieties should be used to further finetune the model. Secondly, the use of average and median (pixel) binning was for reducing the noise presented within the Real-ESRGAN-HR images, leading to enhancement of visualization of coronary lumen for more accurate assessment. However, the pixel binning decreased the spatial resolution of Real-ESRGAN-Average and Real-ESRGAN-Median images by using the average and median values of four pixels to represent one pixel respectively. Since the blooming artifact is caused by using the mean attenuation value of a calcified plaque with high density and a vessel with much lower density to represent these two objects, it was expected that the average binning would have lower performance than the median binning [3]. Our results were in line with this expectation except the aforementioned cases which could be due to presence of mixed calcium and atheromatous plaques. Further finetuning of the Real-ESRGAN model with a greater number and variety of cases should address this issue.”

Section 4: “This study has some limitations. First, although significant improvements in specificity and PPV were achieved over previous studies [3-5,7,27], the diagnostic value of the finetuned Real-ESRGAN-processed images at LAD and LCx is still low to moderate, in particular at the LAD level since it has the largest number of calcified plaques. Further improvement of the Real-ESRGAN model is necessary to address this limitation. Second, as highlighted in our previous study [3], we did not investigate the diagnostic performance of the finetuned Real-ESRGAN model in differentiating calcified from non-calcified plaques as we focused on the heavy calcification in the coronary arteries since this is the main challenging issue to be resolved with CCTA images. Although only 50 patient cases were included in this study but 184 plaques were analyzed which was a greater number than the one of a similar study [7]. Also, for studies about use of AI in radiology, usually, about 50 patient cases were collected for clinical evaluation of the AI models [29]. With improved specificity and PPV, and high AUC achieved with our finetuned model, use of the Real-ESRGAN model is expected to apply to large datasets with inclusion of cases with different types of plaques. Third, these 50 cases were scanned with different types of CT scanners (64-slice and beyond) with sufficient image quality achieved for the diagnostic assessment of coronary plaques. However, the heterogeneity of the original datasets could impact the AI-processed images, thus affecting final outputs of the image assessment. Ideally, CT imaging data from the same type of CT scanners should be used to avoid this issue and this will be addressed in our further study with inclusion of large datasets. Finally, we did not analyze the economic effect as conducted by Inage et al [27]. This will be addressed in future studies when more robust findings are achieved with use of our developed model. It can be expected that further reduction of false positive rates will make significant contribution to reducing ICA procedures in the clinical practice.”

We hope our response provides an adequate clarification for addressing the comment, “Author should add one section “Related Work” in the paper.”

(11) The methodology of the paper should be clearly explained with appropriate flow charts.

Response: Thank you for your comment. A flow chart (Figure 2) showing key steps involved in real-enhanced super-resolution generative adversarial network (Real-ESRGAN) model finetuning and its performance evaluation, and the following content, “… and Figure 1 illustrates the key steps involved in the Real-ESRGAN model finetuning and its performance evaluation” have been added to Section 2.2. for addressing this comment.

(12) Highlight the more applications of the proposed technique.

Response: Thank you for your comment. The following paragraph has been added to Section 4 for addressing this comment.

“Spatial resolution is one of the important elements for visualization of fine details in medical imaging which is essential in accurate diagnosis of various pathological conditions. Hence, the use of Real-ESRGAN model can be extended to other related areas, for example, textual detail restoration for low dose CT images [29], visualization of small soft tissue foreign bodies on digital radiographs [30,31], etc. Nonetheless, the Real-ESRGAN model should be finetuned with relevant medical images before the applications. Another benefit of extending the use of Real-ESRGAN model in these areas is that GAN is less likely affected by the overfitting issue because the generator of the GAN model learns directly from its discriminator’s feedback instead of training / finetuning images. Hence, more robust performance would be expected [32].”

(13) What are motivations behind this research work?

Response: Thank you for your comment. The motivations behind this research work have been stated in Section 1 as follows.

“Coronary artery calcium scoring is widely used in patient screening to enable a more personalized risk assessment [1-3]. However, blooming artifact of coronary computed tomography angiography (CCTA) resulting from extensive calcification within the coronary plaques affects accurate assessment of coronary stenosis, thus leading to high false positive rate. “Blooming” in the calcified plaques refers to partial volume averaging of different densities within a single voxel in the coronary arteries and this is usually caused by limited spatial resolution of computed tomography (CT) scanners. High-density calcium overwhelms the attenuation of other tissues in the voxel and adjacent structures, thus, exaggerates the dimension of the highly calcified plaque (Figure 1). Hence, the high-density calcified plaque appears larger than it is or is “bloomed”, which negatively affects the visualization and assessment of the coronary artery lumen and the degree of stenosis. The consequence of blooming artifact is overestimation of the coronary stenosis which compromises specificity and positive predictive value (PPV) of CCTA, but does not change the sensitivity of CCTA. This leads to unnecessary downstream testing, usually invasive coronary angiography (ICA) which should be avoided in patients without significant coronary stenosis [3-6].”

“One of the main approaches for blooming artifact suppression is to improve the CCTA image spatial resolution. Various strategies for increasing the CCTA image spatial resolution to reduce the blooming artifact have been reported [3-6]. The latest strategy is to use artificial intelligence (AI) (specifically deep learning [DL]) including convolutional neural network (CNN)-based CT image reconstruction kernels such as Canon Medical Systems Advanced Intelligent Clear-IQ Engine (AiCE) and generative adversarial network (GAN) model for image postprocessing to achieve this goal [3,7].”

“Our recent study has shown that enhanced super-resolution GAN (ESRGAN) was able to effectively suppress the CCTA blooming artifact, and improve the specificity and PPV by 10-40% for patients with heavy calcification in the coronary arteries [3]. Its performance was better than the Canon AiCE reconstruction kernel which could only increase the PPV by about 10% [3,7]. Despite of these promising results, AI inference was used in our previous study, i.e. no medical image was used to train the ESRGAN model for the calcium deblooming task [3,8,9]. Hence, one straightforward way to further improve the performance of the ESRGAN model for this task is to finetune the model with use of CCTA images [3,8-11]. The use of finetuning (a subset of transfer learning) has become popular in the medical imaging field because of limited availability of medical images for training a model from scratch, and time- and resource-efficient but still being able to achieve superior performance on similar tasks [12].”

We hope our response provides an adequate clarification for addressing the comment, “What are motivations behind this research work?”

(14) Add more explanation on obtained results with critical analysis.

Response: Thank you for your comment. Explanations of our results with critical analysis have been given in Section 4 as follows.

“This study further advances our recent report of using the finetuned DL model, Real-ESRGAN to postprocess the original CCTA images with results showing significant improvements over previous studies [3-5]. Based on analysis of the same dataset, our results showed that the specificity and PPV were further increased by up to 25% and 15%, respectively compared to our recent results [3], indicating that the finetuned Real-ESRGAN model allows for further improvement in assessing calcified coronary plaques. This has significant clinical impact as the number of false positive rates were reduced, thus reducing the unnecessary downstream testing such as avoiding ICA procedures when diagnosing calcified coronary plaques.”

“Assessment of coronary artery disease, in particular coronary calcification and coronary plaques is a well-recognized issue in CCTA which has drawn increasing attention in recent years to tackle this challenging area. Although a number of strategies have been implemented with some promising results, use of AI algorithm to process original CCTA images represents the most promising strategy in the recent literature [15-23]. Studies have shown that Al algorithm allows for accurate and efficient quantification of coronary calcium scores and assessment of coronary stenosis when compared to the standard manual approach or semi-automatic method [15-19]. Al is also shown to achieve good accuracy in characterizing plaque morphology and differentiating plaque from no plaque, or calcified from non-calcified plaques [20-23]. However, very limited research has been conducted so far with use of AI in suppressing heavy calcification in the coronary arteries for reducing blooming artifacts to improve lumen assessment. Further, most of the previous studies used the traditional CNN model, which is inferior to the advanced GAN approach. Most applications using the GAN approach focus on CT denoising [24], coronary artery disease risk categorization by quantifying calcium scoring [25], and automated registration of positron emission tomography-CT angiography images in imaging coronary artery disease [26]. Our study has addressed this gap by using the latest GAN model with promising results achieved.”

“Inage et al. [27] in their recent study applied the cycle GAN-based lumen extraction model to CCTA images in 99 patients involving 891 segments with severe calcification in the coronary arteries. Diagnostic value of assessing coronary stenosis by the original CCTA and cycle GAN-processed images were compared with ICA as the reference method. In addition to assessing the performance of original CCTA and cycle GAN-processed images in all 891 segments, authors focused on the analysis of 228 segments which were not assessable on the original CCTA images due to severe calcification. Their results showed similar specificity and PPV between the original CCTA and cycle GAN-processed images (75.1% and 40.9% vs. 77.3% and 43.4%) among assessment of all coronary segments, with AUC significantly higher in the cycle GAN group than the original CCTA group (0.77 vs. 0.75, p=0.03). For the non-assessable 228 segments, the cycle GAN model significantly improved the specificity and accuracy compared to the original CCTA (10.9% and 42.5% vs. 0% and 35.5%, p<0.001), along with significantly higher AUC (0.59 vs. 0.50, p<0.001). Similar to assessment of all segments, the PPV was similar between these two groups (35.5% and 38.2% for the original CCTA and cycle GAN). Authors claimed that the use of cycle GAN model could avoid four out of 99 ICA examinations based on their study, with estimated 747 ICA procedures to be avoided per year. Despite promising results achieved within this study, the PPV was low (<45%) with moderate specificity and very low specificity in all segments and non-assessable segments groups. In contrast, our findings showed much better results than those from Inage et al.’s study [27]. Both specificity and PPV were increased significantly with the use of finetuned Real-ESRGAN model, achieving 80% and 61% at the RCA level, although still low at LAD and LCx levels. This represents so far the most promising outcomes of diagnosing calcified plaques. Although we did not evaluate the economic benefits in our study, the improved PPV with reduced false positive rates will lead to avoidance of more unnecessary ICA examinations.”

“With further reduction of false positive rates leading to improved specificity and PPV with the use of finetuned Real-ESRGAN model compared to our previous report [3], the negative effect is the slightly decreased sensitivity due to the false negative rates. Up to three false negative cases were noticed in the Real-ESRGAN-processed images at all of three coronary arteries with the highest number seen in the Real-ESRGAN-Average and Real-ESRGAN-Median groups (Table 1), but not in the original CCTA images. This issue could be attributed to two factors. Firstly, only 32 datasets consisted of 16,904 CCTA images were used to finetune the Real-ESRGAN model. According to Wang et al.’s [11] study about transfer learning of pre-trained GAN models, this arrangement should be sufficient. The unexpected false negative cases would be an indication of the Real-ESRGAN model not exposed to adequate variations of plaque characteristics, e.g. varied compositions with presence of mixed calcium and atheromatous plaques, etc. More cases with greater varieties should be used to further finetune the model. Secondly, the use of average and median (pixel) binning was for reducing the noise presented within the Real-ESRGAN-HR images, leading to enhancement of visualisation-visualization of coronary lumen for more accurate assessment. However, the pixel binning decreased the spatial resolution of Real-ESRGAN-Average and Re-al-ESRGAN-Median images by using the average and median values of four pixels to represent one pixel respectively. Since the blooming artifact is caused by using the mean attenuation value of a calcified plaque with high density and a vessel with much lower density to represent these two objects, it was expected that the average binning would have lower performance than the median binning [3]. Our results were in line with this expectation except the aforementioned cases which could be due to presence of mixed calcium and atheromatous plaques. Further finetuning of the Real-ESRGAN model with a greater number and variety of cases should address this issue.”

“This study has some limitations. First, although significant improvements in specificity and PPV were achieved over previous studies [3-5,7,27], the diagnostic value of the finetuned Real-ESRGAN-processed images at LAD and LCx is still low to moderate, in particular at the LAD level since it has the largest number of calcified plaques. Further improvement of the Real-ESRGAN model is necessary to address this limitation. Second, as highlighted in our previous study [3], we did not investigate the diagnostic performance of the finetuned Real-ESRGAN model in differentiating calcified from non-calcified plaques as we focused on the heavy calcification in the coronary arteries since this is the main challenging issue to be resolved with CCTA images. Although only 50 patient cases were included in this study but 184 plaques were analyzed which was a greater number than the one of a similar study [7]. Also, for studies about use of AI in radiology, usually, about 50 patient cases were collected for clinical evaluation of the AI models [29]. With improved specificity and PPV, and high AUC achieved with our finetuned model, use of the Real-ESRGAN model is expected to apply to large datasets with inclusion of cases with different types of plaques. Third, these 50 cases were scanned with different types of CT scanners (64-slice and beyond) with sufficient image quality achieved for the diagnostic assessment of coronary plaques. However, the heterogeneity of the original datasets could impact the AI-processed images, thus affecting final outputs of the image assessment. Ideally, CT imaging data from the same type of CT scanners should be used to avoid this issue and this will be addressed in our further study with inclusion of large datasets. Finally, we did not analyze the economic effect as conducted by Inage et al [27]. This will be addressed in future studies when more robust findings are achieved with use of our developed model. It can be expected that further reduction of false positive rates will make significant contribution to reducing ICA procedures in the clinical practice.”

We hope our response provides an adequate clarification for addressing your comment. However, if you believe this is insufficient, please specific which part(s) of our results that need(s) further explanation.

(15) Author must explain pros and cons of the work.

Response: Thank you for your comment. The pros and cons of our work have been discussed and explained in Section 4. The details are as follows.

“This study further advances our recent report of using the finetuned DL model, Real-ESRGAN to postprocess the original CCTA images with results showing significant improvements over previous studies [3-5]. Based on analysis of the same dataset, our results showed that the specificity and PPV were further increased by up to 25% and 15%, respectively compared to our recent results [3], indicating that the finetuned Real-ESRGAN model allows for further improvement in assessing calcified coronary plaques. This has significant clinical impact as the number of false positive rates were reduced, thus reducing the unnecessary downstream testing such as avoiding ICA procedures when diagnosing calcified coronary plaques.”

“Inage et al. [27] in their recent study applied the cycle GAN-based lumen extraction model to CCTA images in 99 patients involving 891 segments with severe calcification in the coronary arteries. Diagnostic value of assessing coronary stenosis by the original CCTA and cycle GAN-processed images were compared with ICA as the reference method. In addition to assessing the performance of original CCTA and cycle GAN-processed images in all 891 segments, authors focused on the analysis of 228 segments which were not assessable on the original CCTA images due to severe calcification. Their results showed similar specificity and PPV between the original CCTA and cycle GAN-processed images (75.1% and 40.9% vs. 77.3% and 43.4%) among assessment of all coronary segments, with AUC significantly higher in the cycle GAN group than the original CCTA group (0.77 vs. 0.75, p=0.03). For the non-assessable 228 segments, the cycle GAN model significantly improved the specificity and accuracy compared to the original CCTA (10.9% and 42.5% vs. 0% and 35.5%, p<0.001), along with significantly higher AUC (0.59 vs. 0.50, p<0.001). Similar to assessment of all segments, the PPV was similar between these two groups (35.5% and 38.2% for the original CCTA and cycle GAN). Authors claimed that the use of cycle GAN model could avoid four out of 99 ICA examinations based on their study, with estimated 747 ICA procedures to be avoided per year. Despite promising results achieved within this study, the PPV was low (<45%) with moderate specificity and very low specificity in all segments and non-assessable segments groups. In contrast, our findings showed much better results than those from Inage et al.’s study [27]. Both specificity and PPV were increased significantly with the use of finetuned Real-ESRGAN model, achieving 80% and 61% at the RCA level, although still low at LAD and LCx levels. This represents so far the most promising outcomes of diagnosing calcified plaques.”

“This study has some limitations. First, although significant improvements in specificity and PPV were achieved over previous studies [3-5,7,27], the diagnostic value of the finetuned Real-ESRGAN-processed images at LAD and LCx is still low to moderate, in particular at the LAD level since it has the largest number of calcified plaques. Further improvement of the Real-ESRGAN model is necessary to address this limitation. Second, as highlighted in our previous study [3], we did not investigate the diagnostic performance of the finetuned Real-ESRGAN model in differentiating calcified from non-calcified plaques as we focused on the heavy calcification in the coronary arteries since this is the main challenging issue to be resolved with CCTA images. Although only 50 patient cases were included in this study but 184 plaques were analyzed which was a greater number than the one of a similar study [7]. Also, for studies about use of AI in radiology, usually, about 50 patient cases were collected for clinical evaluation of the AI models [29]. With improved specificity and PPV, and high AUC achieved with our finetuned model, use of the Real-ESRGAN model is expected to apply to large datasets with inclusion of cases with different types of plaques. Third, these 50 cases were scanned with different types of CT scanners (64-slice and beyond) with sufficient image quality achieved for the diagnostic assessment of coronary plaques. However, the heterogeneity of the original datasets could impact the AI-processed images, thus affecting final outputs of the image assessment. Ideally, CT imaging data from the same type of CT scanners should be used to avoid this issue and this will be addressed in our further study with inclusion of large datasets. Finally, we did not analyze the economic effect as conducted by Inage et al [27]. This will be addressed in future studies when more robust findings are achieved with use of our developed model. It can be expected that further reduction of false positive rates will make significant contribution to reducing ICA procedures in the clinical practice.”

We hope our response provides an adequate clarification for addressing the comment, “Author must explain pros and cons of the work.”

(16) What are the major issues in the AI?

Response: Thank you for your comment. The major issues in AI have been discussed in Sections 1 and 4 which are limited availability of medical images for training an AI model from scratch, and number and variety of cases used for model finetuning are still crucial for further performance improvement. Please see below for the details.

Section 1: “The use of finetuning (a subset of transfer learning) has become popular in the medical imaging field because of limited availability of medical images for training a model from scratch, and time- and resource-efficient but still being able to achieve superior performance on similar tasks [12].”

Section 4: “With further reduction of false positive rates leading to improved specificity and PPV with the use of finetuned Real-ESRGAN model compared to our previous report [3], the negative effect is the slightly decreased sensitivity due to the false negative rates. Up to three false negative cases were noticed in the Real-ESRGAN-processed images at all of three coronary arteries with the highest number seen in the Real-ESRGAN-Average and Real-ESRGAN-Median groups (Table 1), but not in the original CCTA images. This issue could be attributed to two factors. Firstly, only 32 datasets consisted of 16,904 CCTA images were used to finetune the Real-ESRGAN model. According to Wang et al.’s [11] study about transfer learning of pre-trained GAN models, this arrangement should be sufficient. The unexpected false negative cases would be an indication of the Real-ESRGAN model not exposed to adequate variations of plaque characteristics, e.g. varied compositions with presence of mixed calcium and atheromatous plaques, etc. More cases with greater varieties should be used to further finetune the model. Secondly, the use of average and median (pixel) binning was for reducing the noise presented within the Real-ESRGAN-HR images, leading to enhancement of visualisation-visualization of coronary lumen for more accurate assessment. However, the pixel binning decreased the spatial resolution of Real-ESRGAN-Average and Real-ESRGAN-Median images by using the average and median values of four pixels to represent one pixel respectively. Since the blooming artifact is caused by using the mean attenuation value of a calcified plaque with high density and a vessel with much lower density to represent these two objects, it was expected that the average binning would have lower performance than the median binning [3]. Our results were in line with this expectation except the aforementioned cases which could be due to presence of mixed calcium and atheromatous plaques. Further finetuning of the Real-ESRGAN model with a greater number and variety of cases should address this issue.”

We hope our response provides an adequate clarification for addressing the comment, “What are the major issues in the AI?”

(17) How over fitting is minimized in the proposed work?

Response: Thank you for your comment. The following sentences have been added to Section 4 for addressing this comment.

“Another benefit of extending the use of Real-ESRGAN model in these areas is that GAN is less likely affected by the overfitting issue because the generator of the GAN model learns directly from its discriminator’s feedback instead of training / finetuning images. Hence, more robust performance would be expected [32].”

(18) Author must cite suggested papers for enhancing the quality of the paper. These are based on related techniques and confusion matrices-

(a) Multi-Feature Fusion Method for Identifying Carotid Artery Vulnerable Plaque

(b) An efficient AR modelling-based electrocardiogram signal analysis for health informatics

(c) Prediction of atherosclerosis pathology in retinal fundal images with machine learning

approaches

(d) 3D Coronary Artery Reconstruction by 2D Motion Compensation Based on Mutual Information

(e) Robust retinal blood vessel segmentation using convolutional neural network and support vector

machine

(f) Real-time estimation of hospital discharge using fuzzy radial basis function network and electronic

health record data

(g) An efficient ALO-based ensemble classification algorithm for medical big data processing

(h) Non-invasive assessment of fractional flow reserve using computational fluid dynamics modelling

from coronary angiography images

(i) Bone metastatic tumour minimisation due to thermal cementoplasty effect, clinical and computational

methodologies

(j) Multiscale Graph Cuts Based Method for Coronary Artery Segmentation in Angiograms

(k) Bio-medical analysis of breast cancer risk detection based on deep neural network

(l) Changes in scale-invariance property of electrocardiogram as a predictor of hypertension

(m) Peak alpha neurofeedback training on cognitive performance in elderly subjects

(n) Modified model for cancer treatment

(o) Assessment of qualitative and quantitative features in coronary artery MRA

(p) A frugal and innovative telemedicine approach for rural India – automated doctor machine

(q) Study of murmurs and their impact on the heart variability

(r) Analysis of salivary components as non-invasive biomarkers for monitoring chronic kidney disease

(s) Coronary three-vessel disease with occlusion of the right coronary artery: What are the most important factors that determine the right territory perfusion?

(t) An improved graph matching algorithm for the spatio-temporal matching of a coronary artery 3D tree sequence

Response: Thank you for suggesting 20 articles to us. We have reviewed these suggestions. Two suggested papers were retracted by the publisher, Springer Nature in 2022 (https://link.springer.com/article/10.1007/s12652-020-02294-3 & https://link.springer.com/article/10.1007/s12652-019-01559-w#citeas). Another 12 articles were published in International Journal of Medical Engineering and Informatics and the other 6 were published in Innovation and Research in BioMedical Engineering. We have checked their relevance to our study by reviewing their abstracts. It seems they are not relevant. The followings are the suggested article citation information, abstracts and our detailed responses about their irrelevance. Hence, we have not included these suggested articles in our paper.

(a) Multi-Feature Fusion Method for Identifying Carotid Artery Vulnerable Plaque

Xu, X.; Huang, L.; Wu, R.; Zhang, W.; Ding, G.; Liu, L.; Chi, M.; Xie, J. Multi-feature fusion method for identifying carotid artery vulnerable plaque, X. Xu, L. Huang, R. Wu et al. IRBM. 2022, 43, 272–227. doi: 10.1016/j.irbm.2021.07.004.

Abstract:

Purpose: Vulnerable plaque of carotid atherosclerosis is prone to rupture, which can easily lead to acute cardiovascular and cerebrovascular accidents. Accurate identification of the vulnerable plaque is a challenging task, especially on limited datasets.

Methods: This paper proposes a multi-feature fusion method to identify high-risk plaque, in which three types of features are combined, i.e. global features of carotid ultrasound images, echo features of regions of interests (ROI) and expert knowledge from ultrasound reports. Due to the fusion of three types of features, more critical features for identifying high-risk plaque are included in the feature set. Therefore, better performance can be achieved even on limited datasets.

Results: From testing all combinations of three types of features, the results showed that the accuracy of using all three types of features is the highest. The experiments also showed that the performance of the proposed method is better than other plaque classification methods and classical Convolutional Neural Networks (CNNs) on the Plaque dataset.

Conclusion: The proposed method helped to build a more complete feature set so that the machine learning models could identify vulnerable plaque more accurately even on datasets with poor quality and small scale.

Response: Based on the article abstract, this article appears irrelevant to our study as this study is related to ultrasound and machine learning was used. Our study was about computed tomography and deep learning.

(b) An efficient AR modelling-based electrocardiogram signal analysis for health informatics

Gupta, V.; Mittal, M.; Mittal, V.; Gupta A. An efficient AR modelling-based electrocardiogram signal analysis for health informatics. International Journal of Medical Engineering and Informatics. 2021, 14, 74-89.

Abstract:

Today, health informatics not only requires correct but also timely diagnosis much before the occurrence of critical stage of the underlying disease. Electrocardiogram (ECG) is one such non-invasive diagnostic tool to establish an efficient computer-aided diagnosis (CAD) system. In this paper, autoregressive (AR) modelling is proposed that is an efficient technique to process ECG signals by estimating its coefficients. In this paper, two parameters viz. atrial tachycardia (AT) and premature atrial contractions (PAC) are considered for evaluating the performance of the proposed methodology for a total of 17 recordings (6 real time and 11 from MIT-BIH arrhythmia database). As compared to K-nearest neighbour (KNN) and principal component analysis (PCA) with AR modelling [also known as Yule-Walker (YW) and Burg method], KNN classifier coupled with Burg method (i.e., Burg + KNN) yielded good results at model order 9. A sensitivity (Se) of 99.95%, specificity (Sp or PPV) of 99.97%, detection error rate (DER) of 0.071%, accuracy (Acc) of 99.93% and mean time discrepancy (MTD) of 0.557 msec are obtained. Consistent higher values of all the performance parameters can lead to the development of an autonomous CAD tool for timely detection of heart diseases as required in health informatics.

Response: Based on the article abstract, this article appears irrelevant to our study as this study is related to electrocardiogram. Our study was about computed tomography.

(c) Prediction of atherosclerosis pathology in retinal fundal images with machine learning

approaches

Parameswari, C.; Siva Ranjani, S. Prediction of atherosclerosis pathology in retinal fundal images with machine learning approaches. J. Ambient Intell. Human Comput. 2021, 12, 6701-6711. doi: 10.1007/s12652-020-02294-3

Response: As per the publisher’s website of this article (https://link.springer.com/article/10.1007/s12652-020-02294-3), this paper was retracted on 4 July 2022. Hence, it appears inappropriate for us to include this article into our paper as a reference.

(d) 3D Coronary Artery Reconstruction by 2D Motion Compensation Based on Mutual Information

Li, S.; Nunes, J.C.; Toumoulin, C.; Luo, L. 3D coronary artery reconstruction by 2D motion compensation based on mutual information. IRBM. 2018, 39, 69-82. doi: 10.1016/j.irbm.2017.11.005.

Abstract:

Background:

3D reconstruction of the coronary arteries can provide more information in the interventional surgery. Motion compensation is one kind of the 3D reconstruction method.

Methods:

We propose a novel and complete 2D motion compensated reconstruction method. The main components include initial reconstruction, forward projection, registration and compensated reconstruction. We apply the mutual information (MI) and rigidity penalty (RP) as registration measure. The advanced adaptive stochastic gradient descent (ASGD) is adopted to optimize this cost function. We generate the maximum forward projection by the simplified distance driven (SDD) projector. The compensated reconstruction adopts the MAP iterative reconstruction algorithm which is based on prior.

Results:

Comparing with the ECG-gating reconstruction and other reference method, the evaluation indicates that the proposed 2D motion compensation leads to a better 3D reconstruction for both the rest and stronger motion phases. The algorithm compensates the residual motion and reduces the artifact largely. As the gating window width increases, the overall image noise decreases and the contrast of the vessels improves.

Conclusions:

The proposed method improved the 3D reconstruction quality and reduced the artifact level. The considerable improvement in the image quality results from motion compensation increases the clinical usability of 3D coronary artery.

Response: Based on the article abstract, this article appears irrelevant to our study as this study is not related to artificial intelligence, machine learning and deep learning. Our study was about deep learning.

(e) Robust retinal blood vessel segmentation using convolutional neural network and support vector

machine

Balasubramanian, K.; Ananthamoorthy, N.P. Robust retinal blood vessel segmentation using convolutional neural network and support vector machine. J. Ambient Intell. Human Comput. 12, 2021, 3559–3569. doi: 10.1007/s12652-019-01559-w.

Response: As per the publisher’s website of this article (https://link.springer.com/article/10.1007/s12652-019-01559-w#citeas), this paper was retracted on 30 May 2022. Hence, it appears inappropriate to include this article into our paper as a reference.

(f) Real-time estimation of hospital discharge using fuzzy radial basis function network and electronic

health record data

Belderrar, A.; Hazzab A. Real-time estimation of hospital discharge using fuzzy radial basis function network and electronic health record data. International Journal of Medical Engineering and Informatics. 2021, 13, 75-83. doi: 10.1504/IJMEI.2021.10033614.

Abstract:

Hospital resources are scarce and should be properly distributed and justified. Information about how long patients stays in critical intensive care units can provide significant benefits to hospital management resources and optimal admission planning. In this paper, we propose an approach for estimating intensive care unit length of stay using fuzzy radial basis function neural network model. The predictive performance of the model is compared to others using data collected over 13,587 admissions and 54 predictive factors from five critical units with discharges between 2001 and 2012. The proposed model compared to others demonstrated higher accuracy and better estimations. The three most influential factors in predicting length of stay at the early stage of pre-admission were demographic characteristics, admission type, and the first location within the hospital prior to critical unit admission. We have found about 63% of patients with multiple chronic conditions, stayed significantly longer in hospital. Enabling the proposed prediction model in clinical decision support system may serve as reference tools for communicating with patients and hospital managers.

Response: Based on the article abstract, this article appears irrelevant to our study as this study is about predication of intensive care unit length of stay. Our study was about blooming artifact reduction for coronary computed tomography angiography.

(g) An efficient ALO-based ensemble classification algorithm for medical big data processing

Ramachandran, S.K.; Manikandan P. An efficient ALO-based ensemble classification algorithm for medical big data processing. International Journal of Medical Engineering and Informatics. 2021, 13, 54-63. doi: 10.1504/IJMEI.2021.10033610.

Abstract:

In this paper, we indented to propose a consolidated feature selection and ensemble-based classification strategy to diminish the medical big data. Here, the proposed system will be the joint execution of both the ant lion optimiser (ALO) and ensemble classifier. So as to limit the impact of an imbalanced healthcare dataset, ALO is used for the optimal feature selection process. The optimised feature sets are classified by utilising the ensemble classification technique. The ensemble classification method uses the diversity of individual classification models to create better classification results. In this paper, the proposed ensemble classification algorithm used the support vector machine (SVM), and recurrent neural network (RNN) classifier and the results of every classifier were consolidated by the majority voting technique. It was watched that the proposed ensemble technique got promising classification accuracy contrasted and other ensemble strategies. This ensemble system can administer datasets, as quick as required giving the imperative help to viably perceive the underrepresented class. The proposed approach will diminish the big medical data precisely and productively. The simulation result shows that the proposed method has better classification when compared with the single classifiers such as random forest (RF), SVM and naïve Bayes classifier.

Response: Based on the article abstract, this article appears irrelevant to our study as this study is just a general application for processing medical big data. However, our study focussed on a specific area in radiology, i.e. blooming artifact reduction for coronary computed tomography angiography.

(h) Non-invasive assessment of fractional flow reserve using computational fluid dynamics modelling

from coronary angiography images

Udaya Kumar, A.; Raghavi, R.; Reshma, R.; Angeline Kirubha, S.P. Non-invasive assessment of fractional flow reserve using computational fluid dynamics modelling from coronary angiography images. International Journal of Medical Engineering and Informatics. 2021, 13, 44-53. doi: 44-53. 10.1504/IJMEI.2021.10023661.

Abstract:

Fractional flow reserve (FFR invasive) is measured by measuring pressure differences across a coronary artery stenosis by coronary catheterisation technique. This is a gold standard method of determining the extent of stenosis. This method has some potential complications such as coronary vessel dissection, embolism, and renal failure. This paper presents a method for an assessment of FFR noninvasively, with coronary CT angiography imaging and fluid dynamics modelling. FFR is calculated as the ratio between pressure distal to stenosis and pressure proximal to stenosis of the coronary artery region segmented from CT angiography image using MIMICS software. ANSYS software is used to determine the FFR (0.73 ± 0.14). FFR can also be computed non-invasively by this technique for other cardiovascular conditions related to peripheral, cerebrovascular and renovascular disease.

Response: Based on the article abstract, this article appears irrelevant to our study as this study is not related to artificial intelligence, machine learning and deep learning. Also, its focus was about fractional flow reserve. Our study was about the use of deep learning for blooming artifact reduction.

(i) Bone metastatic tumour minimisation due to thermal cementoplasty effect, clinical and computational methodologies

Oliveira, V.C.C.; Fonseca, E.M.M.; Belinha, J.; Rua, C.C.; Piloto, P.A.G.; Natal Jorge, R.M. Bone metastatic tumour minimisation due to thermal cementoplasty effect, clinical and computational methodologies. International Journal of Medical Engineering and Informatics. 2021, 13, 35-43. doi: 10.1504/IJMEI.2020.10031214.

Abstract:

The main objective of this work is to study the thermal effect induced by the bone cement polymerisation, in the metastatic tumour minimisation and to understand the role of such procedure in bone tumour necrosis. Different numerical simulations were produced for different cement sizes introduced in a cortical and spongy bone tumour, with or without an intramedullary nail in titanium. The numerical models were built according to average dimensions of patients obtained from digital conventional radiographs. The finite element results allow to conclude about the high temperature spread effect in bone material. In conclusion, values greater than 45°C were obtained in models without the intramedullary nail system. High quantities of cement produce thermal necrosis in bone with more pronounced effect in depth. The temperature located on the intramedullary nail induces heat transfer along the axial length of the bone, due to the metallic nail, justified by its high thermal conductivity.

Response: Based on the article abstract, this article appears irrelevant to our study as this study is about thermal effect. Our study was about the use of deep learning for blooming artifact reduction.

(j) Multiscale Graph Cuts Based Method for Coronary Artery Segmentation in Angiograms

Mabrouk, S.; Oueslati, C.; Ghorbel, F. Multiscale graph cuts based method for coronary artery segmentation in angiograms. IRBM. 2017, 38, 167-175. doi: 10.1016/j.irbm.2017.04.004.

Abstract:

Context:

X-ray angiography is the most used tool by clinician to diagnose the majority of cardiovascular disease and deformations in coronary arteries like stenosis. In most applications involving angiograms interpretation, accurate segmentation is essential to extract the coronary artery tree and thus speed up the medical intervention.

Materials and Methods:

In this paper, we propose a multiscale algorithm based on Graph cuts for vessel extraction. The proposed method introduces the direction information into an adapted energy functional combining the vesselness measure, the geodesic path and the edgeness measure. The direction information allows to guide the segmentation along arteries structures and promote the extraction of relevant vessels. In the multiscale analysis, we study two scales adaptation (local and global). In the local approach, the image is divided into regions and scales are selected within a range including the smallest and largest vessel diameters in each region, while the global approach computes these diameters considering the whole image. Experiments are conducted on three datasets DS1, DS2 and DS3, having different characteristics and the proposed method is compared with four other methods namely fuzzy c-means clustering (FC), hysteresis thresholding (HT), region growing (RG) and accurate quantitative coronary artery segmentation (AQCA).

Results:

Comparing the two proposed scale adaptation, results show that they give similar precision values on DS1 and DS2 and the local adaptation improve the precision on DS3. Standard quantitative measures were used for algorithms evaluation including Dice Similarity measure (DSM), sensitivity and precision. The proposed method outperforms the four considered methods in terms of DSM and sensitivity. The precision values of the proposed method are slightly lower than the AQCA but it remains higher than the three other methods.

Conclusion:

The proposed method in this paper allows to automatically segment coronary arteries in angiography images. A multiscale approach is adopted to introduce the direction information in a graph cuts based method in order to guide this method to better detect curvilinear structures. Quantitative evaluation of the method shows promising segmentation results compared to some segmentation methods from the state-of-the-art.

Response: Based on the article abstract, this article appears irrelevant to our study as this study is about invasive coronary angiography image segmentation. Our study was about blooming artifact reduction for cardiac computed tomography.

(k) Bio-medical analysis of breast cancer risk detection based on deep neural network

Mathappan, N.; Soundariya, R.S.; Natarajan, A.; Gopalan S.K. Bio-medical analysis of breast cancer risk detection based on deep neural network. International Journal of Medical Engineering and Informatics. 2020, 12, 529-541. doi: 10.1504/IJMEI.2020.10032878.

Abstract:

Breast tumour remains a most important reason of cancer fatality among women globally and most of them pass away due to delayed diagnosis. But premature recognition and anticipation can significantly diminish the chances of death. Risk detection of breast cancer is one of the major research areas in bioinformatics. Various experiments have been conceded to examine the fundamental grounds of breast tumour. Alternatively, it has already been verified that early diagnosis of tumour can give the longer survival chance to a patient. This paper aims at finding an efficient set of features for breast tumour prediction using deep learning approaches called restricted Boltzmann machine (RBM). The proposed framework diagnoses and analyses breast tumour patient's data with the help of deep neural network (DNN) classifier using the Wisconsin dataset of UCI machine learning repository and, thereafter assesses their performance in terms of measures like accuracy, precision, recall, F-measure, etc.

Response: Based on the article abstract, this article appears irrelevant to our study as this study is about computer aided diagnosis for breast tumor. Our study was about blooming artifact reduction for cardiac computed tomography.

(l) Changes in scale-invariance property of electrocardiogram as a predictor of hypertension

Helen, M.M.C.; Singh, D.; Deepak, K.K. Changes in scale-invariance property of electrocardiogram as a predictor of hypertension. International Journal of Medical Engineering and Informatics. 2020, 12, 228-236. doi: 10.1504/IJMEI.2020.10028845.

Abstract:

In this study, electrocardiogram signal has been investigated to assess the presence of scale-invariance changes to classify normotensive and hypertensive subject. ECG signal of 20 normotensive and 20 hypertensive subjects is recorded using MP100 system with a sampling rate (fs) of 500 Hz. The scale-invariance changes of ECG signal are analysed using multifractal detrended fluctuation analysis. The width and shape of the multifractal spectrum obtained is used to detect the hypertensive subject. The multifractal spectrum of the normotensive subject exhibit a pure Gaussian behaviour, but for hypertension subject is left truncated. The width of the multifractal spectrum for hypertension subject (1.7158 ± 0.09649) is more as compared to the normotensive subject (1.534 ± 0.23931). Therefore this method can reflect complexity changes of ECG signal during hypertension. Also, the changes in the multifractal behaviour of ECG signal improves the diagnostic tools for calculating hypertension and as a promising diagnostic tool in cardiovascular disease diagnosis and evaluation.

Response: Based on the article abstract, this article appears irrelevant to our study as this study is related to electrocardiogram. Our study was about computed tomography.

(m) Peak alpha neurofeedback training on cognitive performance in elderly subjects

Bobby, J.S. Peak alpha neurofeedback training on cognitive performance in elderly subjects. International Journal of Medical Engineering and Informatics. 2020, 12, 237-247.

Abstract:

Slowing down of thought, memory and thinking is a normal part of aging. Neurofeedback training (NFT) is a relatively new biofeedback technique that focuses on helping a person train themselves to directly affect brain function. In this study, EEG signal was acquired using single channel electrode, amplified using EEG amplifier and connected to the system through data acquisition device (DAQ). The peak alpha band (10-11 Hz) signal was extracted using LabVIEW software. The NFT protocol that was designed presented the neurofeedback training and thereby showed some improvement in their cognitive processing speed. The visual cues were fed to the LabVIEW software by making the subject visualise some animation or hear some audios. Twenty subjects aged between 60 and 65 were considered for this training. This study had investigated whether the training given to the elderly people showed improvement in the cognitive processing speed of their brain activity.

Response: Based on the article abstract, this article appears irrelevant to our study as this study is related to electroencephalogram. Our study was about computed tomography.

(n) Modified model for cancer treatment

Shafigh, M.R. Modified model for cancer treatment. International Journal of Medical Engineering and Informatics. 2020, 12, 248-259.

Abstract:

This paper proposes an optimal method for eradicating cancer, such that it cannot be relapsed. The major issue is that from the dynamical point of view, the tumour free equilibrium point at the end of chemotherapy is still unstable. Mathematically it means that when the chemotherapy is stopped, the dynamic behaviour of the system moves away from the tumour free equilibrium point and the tumour cells starts increasing. To overcome this problem, the stabilisation of the equilibrium point is proposed. According to this method, the vaccine therapy changes the dynamics of the system around the tumour free equilibrium point, and the chemotherapy pushes the system to the domain of attraction of the desired point. It is shown, that, according to the simulation results, after completing the chemotherapy process, the dynamic of the system becomes stable and the cancerous cells converges to zero.

Response: Based on the article abstract, this article appears irrelevant to our study as this study is related to cancer treatment. Our study was about blooming artifact reduction for coronary computed tomography angiography.

(o) Assessment of qualitative and quantitative features in coronary artery MRA

Velut, J.; Lentz, P.A.; Boulmier, D.; Coatrieux, J.L.; Toumoulin, C. Assessment of qualitative and quantitative features in coronary artery MRA. IRBM. 2011, 32, 229-242. doi: 10.1016/j.irbm.2011.05.002.

Abstract:

In this paper, an analysis of the coronary trees using magnetic resonance angiography (MRA) is performed. The objective is to estimate how much MRA is capable to provide insights into the vascular network. A qualitative exploration of the MRA volumes with anatomical labelling by experts is first performed, Quantitative vessel features are then manually extracted providing a ground truth which is further compared to a semi-automatic extraction. This evaluation is carried out on 10 datasets of the SSFP MRA sequence and allows getting a more precise view on the current state-of-the- art as well as on future achievements to be done.

Response: Based on the article abstract, this article appears irrelevant to our study as this study is related to magnetic resonance angiography. Our study was about blooming artifact reduction for coronary computed tomography angiography.

(p) A frugal and innovative telemedicine approach for rural India – automated doctor machine

Aswath, G.I.; Vasudevan, S.K.; Sampath N. A frugal and innovative telemedicine approach for rural India – automated doctor machine. International Journal of Medical Engineering and Informatics. 2020, 12, 278-290. doi: 10.1504/IJMEI.2020.10028848.

Abstract:

Rural India is very poorly connected when compared to urban. Most of the developments have not yet reached rural India and most importantly, in healthcare, it lacks good hospitals and sometimes they even don't have a good dispensary. Even when there are hospitals, rural people never get experienced doctors. We know that rural people are more affected by many challenging diseases every year. To compensate all these problems and to provide a good healthcare even in rural villages we introduce an automated doctor machine. This machine can be installed easily anywhere and also include all the required input sensors inbuilt to diagnose the condition of the patient and also provide a diagnosis report with medicines. The patient's health report is updated in the health record database in his cloud account once he receives the medicine. So, when he meets the doctor in future, the doctor can easily get his up to date health record and can also use the web app to update his health record. This is all frugal, cost-effective and fault tolerant.

Response: Based on the article abstract, this article appears irrelevant to our study as this study is related to telemedicine. Our study was about blooming artifact reduction for coronary computed tomography angiography.

(q) Study of murmurs and their impact on the heart variability

Mokeddem, F.; Meziani, F.; Debbal, S.M. Study of murmurs and their impact on the heart variability. International Journal of Medical Engineering and Informatics.  2020, 12, 291-301.

Abstract:

The phonocardiogram (PCG) signal processing approach seems to be very revealing for the diagnosis of pathologies affecting the activity of the heart. The aim of this paper is the application of an algorithm to explore heart sounds with simplicity in order to provide statistical parameters to better understand the cardiac activity by the calculation of the cardiac frequency and examine the impact of minor and pronounced murmurs on the cardiac variability. This paper is conducted to clarify the variation of the cardiac frequency affected by several groups of murmurs and investigate the medical reasons behind the obtained results. A high number of cycles were used to better refine the expected results.

Response: Based on the article abstract, this article appears irrelevant to our study as this study is related to phonocardiogram. Our study was about blooming artifact reduction for coronary computed tomography angiography.

(r) Analysis of salivary components as non-invasive biomarkers for monitoring chronic kidney disease

Bhaskar, N.; Suchetha, M. Analysis of salivary components as non-invasive biomarkers for monitoring chronic kidney disease. International Journal of Medical Engineering and Informatics. 2020, 12, 95-107. doi: 10.1504/IJMEI.2020.10020254.

Abstract:

Saliva, a valuable source of biochemical information, is a potential diagnostic substance that helps to identify many diseases. Studies have revealed that saliva tests help identify many diseases. Saliva test has excellent advantages over blood test as the former can be collected non-invasively using simple equipment. This paper explores how salivary components can be used as diagnostic tool to identify chronic kidney disease (CKD). Experimental analysis was conducted to assess the levels of salivary components in whole saliva of CKD patients in contrast with healthy people. Urea and creatinine are the most accepted biomarkers of CKD. The correlation between creatinine and urea levels in human saliva and blood were analysed. Unstimulated saliva flow rate and pH levels were also monitored in this study. The results obtained from this study give concrete evidence that there is a positive correlation between creatinine and urea levels in blood and saliva. From the derived regression line equations, serum urea and creatinine values can be predicted from salivary values. Receiver operating characteristics (ROC) performance analysis was performed and area under the curve (AUC) of 0.95 and 0.89 was obtained for salivary creatinine and urea.

Response: Based on the article abstract, this article appears irrelevant to our study as this study is related to saliva analysis. Our study was about blooming artifact reduction for coronary computed tomography angiography.

(s) Coronary three-vessel disease with occlusion of the right coronary artery: What are the most important factors that determine the right territory perfusion?

Harmouche, M.; Maasrani, M.; Verhoye, J.P.; Corbineau, H., Drochon, A. Coronary three-vessel disease with occlusion of the right coronary artery: What are the most important factors that determine the right territory perfusion? IRBM. 2014, 35, 149-157. doi: 10.1016/j.irbm.2013.11.002.

Abstract:

With progressive occlusion of a coronary main artery, some anastomotic vessels are recruited in order to supply blood to the ischemic region. This collateral circulation is an important factor in the preservation of the myocardium until reperfusion of the area at risk. An accurate estimation of collateral flow is crucial in surgical bypass planning as it alters the blood flow distribution in the coronary network and can influence the outcome of a given treatment for a given patient. The evaluation of collateral flow is frequently achieved using an index based on pressure measurements. It is named collateral flow index (CFI) and defined as: (Pw − Pv)/(Pao − Pv), where Pw is the pressure distal to the thrombosis, Pao the aortic pressure and Pv the central venous pressure. In the present work, we study patients with severe coronary disease (stenoses on the left branches and total occlusion of the right coronary artery). Using a mathematical model that describes the coronary hemodynamics in that situation, we demonstrate that the dependence of the collateral circulation to the pressure values is not as simple as it is commonly believed: using pressures alone as an index of collateral flow is likely to result in misinterpretation of the collateral flow contribution, because collateral flow depends on many other factors related to the status of the native stenosed arteries and to the microvascular resistances (capillary and collateral resistances, and the proportion between them).

Response: Based on the article abstract, this article appears irrelevant to our study as this study is related to flow analysis in invasive coronary angiography. Our study was about blooming artifact reduction for cardiac computed tomography.

(t) An improved graph matching algorithm for the spatio-temporal matching of a coronary artery 3D tree sequence

Feuillâtre, H.; Nunes, J.C.; Toumoulin, C. An improved graph matching algorithm for the spatio-temporal matching of a coronary artery 3D tree sequence. IRBM. 2015, 36, 329-334. doi: 10.1016/j.irbm.2015.09.002.

Abstract:

The paper describes an inexact tree-matching algorithm to register non-isomorphic 3D coronary artery trees over time. This work is carried out in the frame of the determination of the optimal viewing angles on the C-arm acquisition system for coronary percutaneous procedure. The matching method is based on association graph and maximum clique. Different similarity measures are compared, which use tree characteristics and geometric features of vascular branches. In order to take into account the topology variation between 3D vascular trees and thus improve the performance of the algorithm, we propose to insert artificial nodes in the association graph. Results show that unmatched node rate significantly decreases with the insertion of artificial nodes.

Response: Based on the article abstract, this article appears irrelevant to our study as this study is related to invasive coronary angiography. Our study was about blooming artifact reduction for cardiac computed tomography.

Reviewer 2 Report

The paper describes an improved imaging method for calcium de-blooming of CCTA images.  The authors have been involved in this area of research for a number of years.  Early this year, they published a paper in Diagnositics (Ref 3) on the similar topic (i.e. an improved imaging method), and the submitted work is an extension of this study.  As such, the overall progress made by the study is incremental.  This work was submitted to the special issue on Personalized Medicine.  There is nothing that meets this criteria.  It would help if the authors could comment on how this study relates to personalized medicine in the Introduction or Discussion.  I have only minor specific comments.

1.  It would be helpful if the calcium blooming issue is briefly discussed in the Introduction, describing how it happens and its potential clinical problems.

2.  My brief inspection appears to indicate that the blue line in Fig. 3B looks identical to the blue line in Fig. 3B of the earlier paper (Ref. 3).

3.  What are the blue dots in Fig. 1? 

Author Response

The paper describes an improved imaging method for calcium de-blooming of CCTA images.  The authors have been involved in this area of research for a number of years.  Early this year, they published a paper in Diagnositics (Ref 3) on the similar topic (i.e. an improved imaging method), and the submitted work is an extension of this study.  As such, the overall progress made by the study is incremental.  This work was submitted to the special issue on Personalized Medicine. There is nothing that meets this criteria. It would help if the authors could comment on how this study relates to personalized medicine in the Introduction or Discussion.  I have only minor specific comments.

Response: Thank you for your comment. The following paragraph, “One of the important themes in modern health care is personalized medicine which generally refers to tailoring service delivery based on patient’s conditions. Our finetuned Real-ESRGAN model significantly improves the diagnostic performance of CCTA in assessing calcified plaques which is one of the causes of coronary artery stenosis. Hence, our work advances the development of personalized medicine by using the latest DL technology to provide a better diagnostic service to a specific group of patients with the coronary artery stenosis caused by the calcified plaques [28].” has been added to the Discussion section for addressing this comment.

  1. It would be helpful if the calcium blooming issue is briefly discussed in the Introduction, describing how it happens and its potential clinical problems.

Response: Thank you for your suggestion. The following discussion of the calcium blooming issue and its associated clinical problem, “However, blooming artifact of coronary computed tomography angiography (CCTA) resulting from extensive calcification within the coronary plaques affects accurate assessment of coronary stenosis, thus leading to high false positive rate. “Blooming” in the calcified plaques refers to partial volume averaging of different densities within a single voxel in the coronary arteries and this is usually caused by limited spatial resolution of computed tomography (CT) scanners. High-density calcium overwhelms the attenuation of other tissues in the voxel and adjacent structures, thus, exaggerates the dimension of the highly calcified plaque (Figure 1). Hence, the high-density calcified plaque appears larger than it is or is “bloomed”, which negatively affects the visualization and assessment of the coronary artery lumen and the degree of stenosis. The consequence of blooming artifact is overestimation of the coronary stenosis which compromises specificity and positive predictive value (PPV) of CCTA, but does not change the sensitivity of CCTA. This leads to unnecessary downstream testing, usually invasive coronary angiography (ICA) which should be avoided in patients without significant coronary stenosis [3-6].” has been added into the Introduction for addressing this comment.

  1. My brief inspection appears to indicate that the blue line in Fig. 3B looks identical to the blue line in Fig. 3B of the earlier paper (Ref. 3).

Response: Yes, you are right about it. Since the same datasets were used as in our previous paper (ref 3), the diagnostic value of original CCTA remains unchanged.

  1. What are the blue dots in Fig. 1?

Response: The blue dots indicate outliners and this has been explained in the figure legend.

Round 2

Reviewer 1 Report

Paper is important for health informatics. Following revisions are to be incorporated before publication-

(1)   Author should add the motivations, problem, and solution statement in the abstract.

(2)   How the parameters for simulations are selected?

(3)   How the performance of proposed technique is better than existing techniques.

(4)   All tables and figures should be explained clearly.

(5)   The English and typo errors of the paper should be checked in the presence of native English speaker.

(6)   All equations should be clearly explained with explanation on all associated variables.

(7)   Author should add one section “Related Work” in the paper.

(8)   The methodology of the paper should be clearly explained with appropriate flow charts.

(9)   Highlight the more applications of the proposed technique.

(10) How data sets are established for doing this research work?

(11) Which files of datasets are using in this paper? Mention these websites.

(12) What are the strong features of this research work? Author must explain.

(13) What are motivations behind this research work?

(14) Add more explanation on obtained results with critical analysis.

(15) Author must explain pros and cons of the work.

(16) What are the major issues in the AI?

(17) How over fitting is minimized in the proposed work?

(18) Author must cite suggested papers for enhancing the quality of the paper. These are based on realted techniques and confusion matrices-

(a)    Multi-Feature Fusion Method for Identifying Carotid Artery Vulnerable Plaque

(b)   An efficient AR modelling-based electrocardiogram signal analysis for health informatics

(c)    Prediction of atherosclerosis pathology in retinal fundal images with machine learning approaches

(d)   3D Coronary Artery Reconstruction by 2D Motion Compensation Based on Mutual Information

(e)    Robust retinal blood vessel segmentation using convolutional neural network and support vector machine

(f)    Real-time estimation of hospital discharge using fuzzy radial basis function network and electronic health record data

(g)   An efficient ALO-based ensemble classification algorithm for medical big data processing

(h)   Non-invasive assessment of fractional flow reserve using computational fluid dynamics modelling from coronary angiography images

(i)    Bone metastatic tumourminimisation due to thermal cementoplasty effect, clinical and computational methodologies

(j)    Multiscale Graph Cuts Based Method for Coronary Artery Segmentation in Angiograms

(k)   Bio-medical analysis of breast cancer risk detection based on deep neural network

(l)    Changes in scale-invariance property of electrocardiogram as a predictor of hypertension

(m)  Peak alpha neurofeedback training on cognitive performance in elderly subjects

(n)   Modified model for cancer treatment

(o)   Assessment of qualitative and quantitative features in coronary artery MRA

(p)   A frugal and innovative telemedicine approach for rural India – automated doctor machine

(q)   Study of murmurs and their impact on the heart variability

(r)    Analysis of salivary components as non-invasive biomarkers for monitoring chronic kidney disease

(s)    Coronary three-vessel disease with occlusion of the right coronary artery: What are the most important factors that determine the right territory perfusion?

(t)    An improved graph matching algorithm for the spatio-temporal matching of a coronary artery 3D tree sequence
